# Three-dimensional biofilm colony growth supports a mutualism involving matrix and nutrient sharing

**Heidi A Arjes[1], Lisa Willis[1], Haiwen Gui[1], Yangbo Xiao[1], Jason Peters[2,3,4,5,6], Carol Gross[2], Kerwyn Casey Huang[1,7,8]\***

[1]Department of Bioengineering, Stanford University School of Medicine, Stanford, United States; [2]Department of Cell and Tissue Biology, University of California San Francisco, San Francisco, United States; [3]Pharmaceutical Sciences Division, School of Pharmacy, University of Wisconsin-Madison, Madison, United States; [4]Great Lakes Bioenergy Research Center, Wisconsin Energy Institute, University of Wisconsin-Madison, Madison, United States; [5]Department of Bacteriology, University of Wisconsin-Madison, Madison, United States; [6]Department of Medical Microbiology and Immunology, University of Wisconsin-Madison, Madison, United States; [7]Department of Microbiology & Immunology, Stanford University School of Medicine, Stanford, United States; [8]Chan Zuckerberg Biohub, San Francisco, United States

**Abstract** Life in a three-dimensional biofilm is typical for many bacteria, yet little is known about how strains interact in this context. Here, we created essential gene CRISPR interference knockdown libraries in biofilm-forming *Bacillus subtilis* and measured competitive fitness during colony co-culture with wild type. Partial knockdown of some translation-related genes reduced growth rates and led to out-competition. Media composition led some knockdowns to compete differentially as biofilm versus non-biofilm colonies. Cells depleted for the alanine racemase AlrA died in monoculture but survived in a biofilm colony co-culture via nutrient sharing. Rescue was enhanced in biofilm colony co-culture with a matrix-deficient parent due to a mutualism involving nutrient and matrix sharing. We identified several examples of mutualism involving matrix sharing that occurred in three-dimensional biofilm colonies but not when cultured in two dimensions. Thus, growth in a three-dimensional colony can promote genetic diversity through sharing of secreted factors and may drive evolution of mutualistic behavior.

**\*For correspondence:**
kchuang@stanford.edu

**Competing interests:** The authors declare that no competing interests exist.

## Introduction

In natural environments, many bacteria grow in dense, three-dimensional multicellular communities held together by extracellular matrix, often called biofilms. Biofilms have widespread clinical (*Wu et al., 2015*), industrial (*Lappin-Scott and Costerton, 1989*), and biotechnological (*Morikawa, 2006*) implications. Biofilms allow for genetic differentiation and division of labor that can mutually benefit distinct genotypes. For instance, in a dual-species biofilm, extracellular matrix components were functionally exploited by multiple species to drive emergent structural and mechanical properties of the biofilm that affected viability (*Yannarell et al., 2019*). Additionally, the amount of mutations that are supported in a population of a given size is higher in microbial colonies compared to well-shaken, liquid cultures that allow beneficial mutations to sweep the population (*Fusco et al., 2016*; *Habets et al., 2007*), suggesting that spatial confinement supports an increase in genetic variation. Spatial confinement dramatically increases the frequency of interactions between nearby cells

and thus the potential for linked evolutionary outcomes, coupling random genetic drift events in these subpopulations (*Kayser et al., 2018*). However, mechanisms that support genetic diversity in the context of a three-dimensional bacterial colony or biofilm remain underexplored (*Kim et al., 2016*).

The model organism *Bacillus subtilis* is a soil-dwelling species that adheres to plant roots as a biofilm (*Beauregard et al., 2013*). In laboratory conditions, *B. subtilis* grows on surfaces in biofilm or non-biofilm colonies depending on the growth medium. In the commonly used rich medium lysogeny broth (LB), *B. subtilis* grows as a colony with limited biofilm characteristics (*Shemesh and Chai, 2013*). By contrast, when cultured on the biofilm-promoting minimal medium MSgg (*Branda et al., 2001*), *B. subtilis* natural isolates produce extracellular matrix composed of secreted polysaccharides and proteinaceous components that hold cells together and enhance biofilm colony expansion (*Kim et al., 2016*; *Seminara et al., 2012*; *van Gestel et al., 2015*). The extracellular matrix also establishes colony architecture through the three-dimensional pattern of growth and wrinkling (*Asally et al., 2012*). This architecture creates a variety of contexts for genetically identical cells to differentially express genes depending on their location, and indeed biofilms contain functionally distinct subpopulations (*López et al., 2010*; *Vlamakis et al., 2013*): living cells differentiate into extracellular matrix producers, sporulating cells, and motile cells, while dead cells may be cannibalized (*Blair et al., 2008*; *Cairns et al., 2014*; *Vlamakis et al., 2008*). Thus, biofilms are an environment with heightened potential for interactions among cells in distinct transcriptional states and/or genetic backgrounds. Furthermore, biofilm-specific interactions can be identified and characterized by comparing biofilm and non-biofilm growth conditions.

The explosion of interest in microbial communities in recent years has stimulated a variety of approaches for identifying interspecies interactions. Liquid co-cultures have been used to quantify interaction networks (*Venturelli et al., 2018*) and dissect changes in antibiotic sensitivity in co-cultures (*Aranda-Díaz et al., 2020*), but liquid growth cannot be used to identify biofilm or colony-specific interactions as it removes the spatial context of community growth and likely prioritizes long-range interactions over short-range physical contacts. Moreover, determining the amount of each strain in a co-culture often relies on laborious methods such as dilution plating and colony counting (*Aranda-Díaz et al., 2020*; *Arjes et al., 2020*), which may be complicated if cells adhere to each other. Microfluidics has facilitated the production and high-throughput analysis of picoliter (*Ohan et al., 2019*) and nanoliter (*Kehe et al., 2019*) droplets with mixed species, but these approaches again rely on liquid growth and the resulting communities contain very few cells compared to many natural communities, especially those that grow in colonies adhered to a surface, making it difficult to study complex fitness phenotypes beyond those affecting initial growth. The production of antibiotics by certain species has been investigated using the inhibition of colony growth of other species at a distance (*Stubbendieck et al., 2016*; *Temkin et al., 2019*; *Zhang and Straight, 2019*), and a colony-based screen identified interspecies interactions between *B. subtilis* and other soil bacteria (*Shank et al., 2011*). While powerful, these methods are not amenable to investigating genetically distinct strains growing together in a three-dimensional structure. The increased availability of mutant libraries across organisms (*Baba et al., 2006*; *Koo et al., 2017*; *Peters et al., 2016*; *Rousset et al., 2018*) motivates the development of a colony-based strategy for high-throughput screening of the fitness of strains within co-cultures.

Both non-essential and essential genes (so defined based on survival in a typical laboratory environment such as liquid growth in LB) may impact fitness in any environment. While chemical genetic screens of ordered libraries of deletions of non-essential genes have revealed novel phenotypes and elucidated the mechanism of action of drugs (*Brochado and Typas, 2013*; *Nichols et al., 2011*), and phenotypic screens of transposon libraries have identified sporulation- and biofilm-related non-essential genes (*Branda et al., 2004*; *Meeske et al., 2016*), essential genes have traditionally been challenging to address. CRISPR interference (CRISPRi) utilizes an endonuclease-dead version of Cas9 (dCas9) to inhibit transcription from a gene of interest (*Qi et al., 2013*), facilitating tunable expression of any gene. Previously, we created a library of CRISPRi knockdowns of each essential gene in the non-biofilm-forming strain *B. subtilis* 168, which we used to uncover essential gene networks and identify functional classes of genes based on growth and morphology (*Peters et al., 2016*). In each strain of this library, the level of an essential gene can be titrated, from basal knockdown that allows robust growth of cells in liquid cultures, to full knockdown that inhibits the growth of many strains (*Peters et al., 2016*). Thus, CRISPRi targeting of essential genes provides the potential for a wide

distribution of phenotypes, enabling determination of the effects of essential gene disruption with-out completely inhibiting growth (*Larson et al., 2013*). This ability to achieve tunable knockdown is particularly appealing for quantifying interstrain interactions, by contrast to the lethal phenotype of complete removal of essential genes. CRISPRi was recently used to identify genes that regulate bio-film formation in *Pseudomonas fluorescens* (*Noirot-Gros et al., 2019*); the efficacy of CRISPRi for *B. subtilis* in a colony/biofilm environment has yet to be ascertained.

Here, we created Green Fluorescent Protein (GFP)-labeled libraries of CRISPRi essential gene knockdowns in the biofilm-forming strain *B. subtilis* NCIB 3610 to investigate the fitness consequences of gene knockdowns and interstrain interactions within three-dimensional biofilm and non-biofilm colonies. We demonstrated that the level of CRISPRi knockdown is tunable during colony growth on LB and biofilm-promoting MSgg agar. We developed a high-throughput method for screening monocultures and co-culture colonies on agar plates, and applied this method to quantify growth and fitness when CRISPRi knockdowns were co-cultured with a wild-type-like parent strain. We observed a wide range of fitness phenotypes across media and knockdown levels, with partial knockdowns of translation-related genes producing the lowest fitness, likely due to their negative impact on growth rate. We discovered that full knockdown of *alrA*, which encodes an alanine race-mase required for cell wall synthesis, was rescued by the presence of wild-type cells in a co-culture biofilm colony but not in liquid. This rescue was enhanced and stable over time when parent cells were unable to produce extracellular matrix, revealing a mutualistic interaction between these strains. Finally, we identified several other knockdowns with higher competitive fitness when the parent cells are deficient in extracellular matrix production as long as biofilm colony growth occurs in three dimensions, suggesting that these genes have mutualistic potential via nutrient and matrix sharing. These findings highlight the importance of colony geometry and matrix production in determining gene essentiality and interstrain genetic interactions, and provide foundational knowledge of mechanisms that support genetic diversity in pathogenic and environmental biofilms.

## Results

### Construction of a knockdown library for probing gene essentiality in *B. subtilis* 3610

To study genetic interactions involving critical cellular processes within a biofilm colony, we constructed a CRISPRi knockdown library in the biofilm-forming *B. subtilis* strain 3610 (Materials and methods). The library contains 302 strains: the 252 known essential genes in *B. subtilis* strain 168, 47 genes that were initially classified as essential in 168 (*Kobayashi et al., 2003*) but later revealed to be non-essential or conditionally essential (*Koo et al., 2017*), and three internal controls expressing dCas9 without any guide RNAs (*Supplementary file 1*; *Peters et al., 2016*). Each strain in the library contains a xylose-inducible copy of *dcas9* and an sgRNA targeting the gene of interest (*Figure 1A*). In addition, *gfp* is incorporated at the *sacA* locus to allow visualization and quantification of the knockdown strain (*Figure 1A*). The *sacA::gfp* strain exhibited similar colony growth and biofilm wrinkling as a parental unlabeled control on both non-biofilm LB agar and biofilm-promoting MSgg (*Branda et al., 2001*) agar (*Figure 1B*). We refer to colonies on LB and MSgg as 'non-biofilm' colonies and 'biofilm' colonies, respectively. However, it is important to note that the biofilm definition is nuanced; colonies on LB may have some biofilm characteristics (*Shemesh and Chai, 2013*), and biofilm pellicles formed at air–liquid interfaces may have distinct properties from MSgg colonies.

To determine whether CRISPRi can be used to knock down gene expression in non-biofilm and biofilm colonies, we engineered a parent strain containing the Red Fluorescent Protein gene (*rfp*) under a constitutive promoter and used CRISPRi to target *rfp*. In this strain, RFP levels in non-biofilm colonies on LB agar plates ranged from 40% (basal knockdown) to ~0% (full knockdown) (*Figure 1—figure supplement 1A*), a comparable range to knockdown of the domesticated strain 168 in liquid LB (*Peters et al., 2016*). RFP levels in biofilm colonies on MSgg agar ranged from ~90% (basal knockdown) to ~0% (full knockdown) (*Figure 1—figure supplement 1A*). Thus, CRISPRi is an effective tool to knock down gene expression in non-biofilm and biofilm colonies.

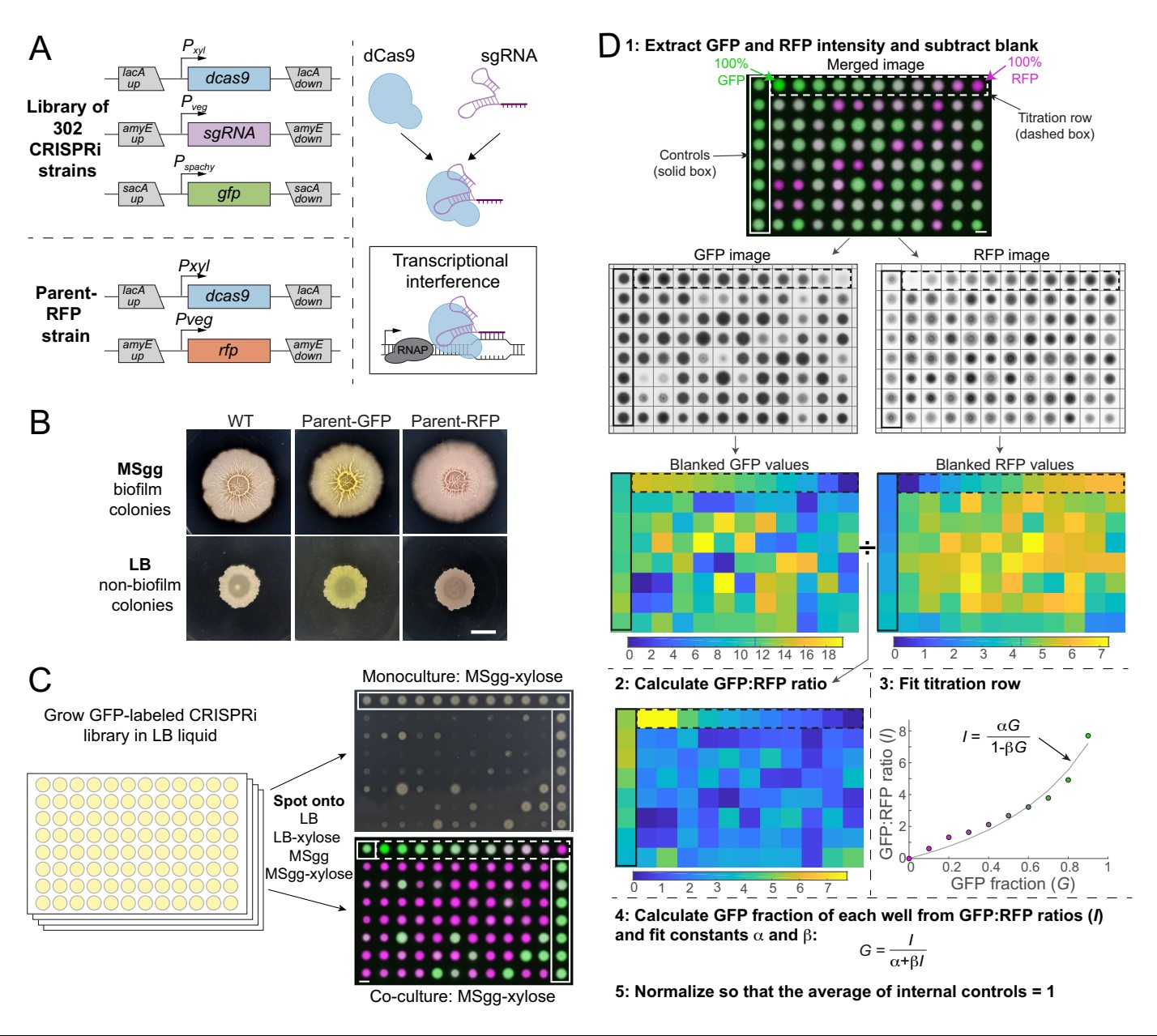

**Figure 1.** A high-throughput screening strategy to measure colony-based competition within bacterial colonies. (**A**) We constructed a GFP-labeled library of CRISPR interference (CRISPRi) knockdowns of all known essential and conditionally essential genes (top left). In the library, the nuclease-deactivated Cas9 gene (*dcas9*) is inducible with xylose and the single-guide RNA (*sgRNA*) is constitutively expressed. dCas9 binds the sgRNA and blocks transcription by physically impeding RNA polymerase (right). Every strain is labeled with *gfp* expressed from the *sacA* locus. A parent strain (parent-RFP, bottom left) that expresses *rfp* as well as *dCas9* without an *sgRNA* was used in competition assays. (**B**) The parent-GFP (*sacA::gfp*, *lacA:: dCas9*) and parent-RFP strains have similar phenotypes to wild type on both biofilm-promoting MSgg agar and non-biofilm-promoting LB agar. Cultures were grown in liquid LB to an optical density (OD$_{600}$) ~1, and then 1 μL was spotted in the middle of wells of a six-well plate containing LB agar or MSgg agar. Image intensities were adjusted identically; the yellow and red colors of the parent strains are due to GFP and RFP expression, respectively. Scale bar: 5 mm. (**C**) Schematic of screening strategy to measure the monoculture colony size and competitive fitness of each knockdown against the parent-RFP strain. GFP-labeled knockdown libraries were grown in liquid culture in 96-well microtiter plates. Monocultures were spotted onto LB and MSgg agar plates (top right) without or with xylose to achieve basal or full knockdown, respectively, of the targeted gene. The monoculture plates contained parent controls in wells along an outer column and row (solid box). Co-cultures of a 1:1 volumetric mixture of the parent-RFP and GFP-labeled library strains were spotted onto agar plates of LB and MSgg, without or with xylose. Controls in which parent-GFP was mixed with parent-RFP are bounded by horizontal red box. Bottom right: merged image of RFP and GFP signals from co-cultures. The co-cultures include a titration row from 100% GFP cultures to 100% RFP cultures in 10% increments (dashed box), and several controls of 1:1 mixtures of the parent GFP and

*Figure 1 continued on next page*

*Figure 1 continued*

parent-RFP strains (purple box). Scale bar: 5 mm. (**D**) Schematic of image analysis to quantify competitive fitnesses from the co-culture screen. Data from plate 1 spotted on MSgg is presented as an example. Plates were segmented and individual colony intensities were extracted from the GFP and RFP images. GFP intensities were divided by RFP intensities to obtain ratios *I*. The titration row (dashed box) was fit to a curve using the equation $I=\alpha G/(1-\beta G)$, where $G$ is the fraction of the parent-GFP strain, to extract fit parameters $\alpha$ and $\beta$ for each plate individually. These parameters were used to map the GFP fractions of each colony and values were normalized so that the parent-GFP:parent-RFP control co-cultures on each plate (solid box) had an average value of 1. Scale bar 5 mm.

The online version of this article includes the following figure supplement(s) for figure 1:

**Figure supplement 1.** CRISPRi is an efficient tool for tunable knockdown of gene expression in non-biofilm and biofilm colonies, enabling high-throughput competition screens.

**Figure supplement 2.** Images of plates from the competition screen with the *sacA::GFP* library.

## High-throughput screening of competitive fitness in a colony

We compared the colony growth phenotypes of GFP-labeled knockdown strains grown either alone or mixed with a control strain modified with xylose-inducible dCas9 (without an sgRNA) and constitutive expression of RFP (henceforth referred to as parent-RFP) that exhibits wild-type-like biofilm colony formation (*Figure 1A, B*). After growing each strain individually in liquid LB, GFP-labeled knockdown strains were spotted either alone or mixed with parent-RFP onto agar plates (*Figure 1C*, *Figure 1—figure supplement 2*). Each plate included a titration row of colonies grown from mixtures of a parent-GFP strain with the parent-RFP strain at known concentrations from 0% to 100% parent-GFP (*Figure 1D*, *Figure 1—figure supplement 2*). Quantification of the titration row closely agreed with the predicted ratio of GFP:RFP at each time point (16, 24, and 48 hr) (*Figure 1D*, S1B, E), indicating that the relative fraction of GFP-labeled mutants in co-culture with the parent-RFP strain can be accurately quantified through comparison of the GFP:RFP ratio with the titration row (Materials and methods). The titration row also led to good agreement with the predicted ratio of GFP:RFP when LB and MSgg colonies were disrupted by sonication and plated to determine relative colony-forming units (*Figure 1—figure supplement 1B*), further validating our fluorescence-based assay. Colony phenotypes were quantified using a custom image-analysis pipeline that segmented plates into colonies and computed the ratio of GFP:RFP for each colony; colony size was measured manually (*Figure 1D*, S1C, D; Materials and methods). Thus, our screen allows us to quantitatively compare growth as a monoculture to growth in co-culture through this competitive fitness value (*Figure 1C*).

## Gene knockdown results in a broad range of colony sizes and competitive fitnesses

To measure growth in monocultures or co-cultures across conditions, we spotted knockdowns on their own or mixed with parent-RFP on LB and MSgg agar without and with xylose. After 16 hr of growth, colony sizes of basal knockdown monocultures exhibited a narrow distribution on LB agar, but were more widely distributed on MSgg agar (*Figure 2A*, *Figure 2—figure supplement 1A*, *Supplementary file 2*), indicating increased sensitivity to knockdown on MSgg. By contrast to growth as monocultures on LB, basal knockdowns co-cultured with the parent for 16 hr exhibited a broad distribution of competitive fitness on LB agar: only 92 of the strains had a fitness within two standard deviations of the mean of controls, while the remaining 210 strains were significantly defective in competition (below two standard deviations of the mean of the controls) (*Figure 2B*, *Supplementary file 2*). A competitive fitness of 1 signifies equal amounts of GFP-labeled knockdown and parent-RFP, and 0 means that the GFP-labeled knockdown was completely outcompeted by the parent-RFP strain. On MSgg agar, competitive fitness displayed a similar trend, with 198 basal knockdowns exhibiting a significant fitness defect after 16 hr (*Figure 2A, B*, *Supplementary file 2*). As expected, when transcription was fully knocked down, fitness was even more compromised: 168 and 143 strains had fitness <0.08 after 16 hr of growth on LB xylose and MSgg xylose agar, respectively (*Figure 2B*, *Supplementary file 2*). Together, these data demonstrate that even though phenotypes were generally subtle for monocultures grown on non-biofilm-promoting LB agar, screening the library on biofilm-promoting MSgg agar or in competition with a parent strain uncovered phenotypes even under basal knockdown.

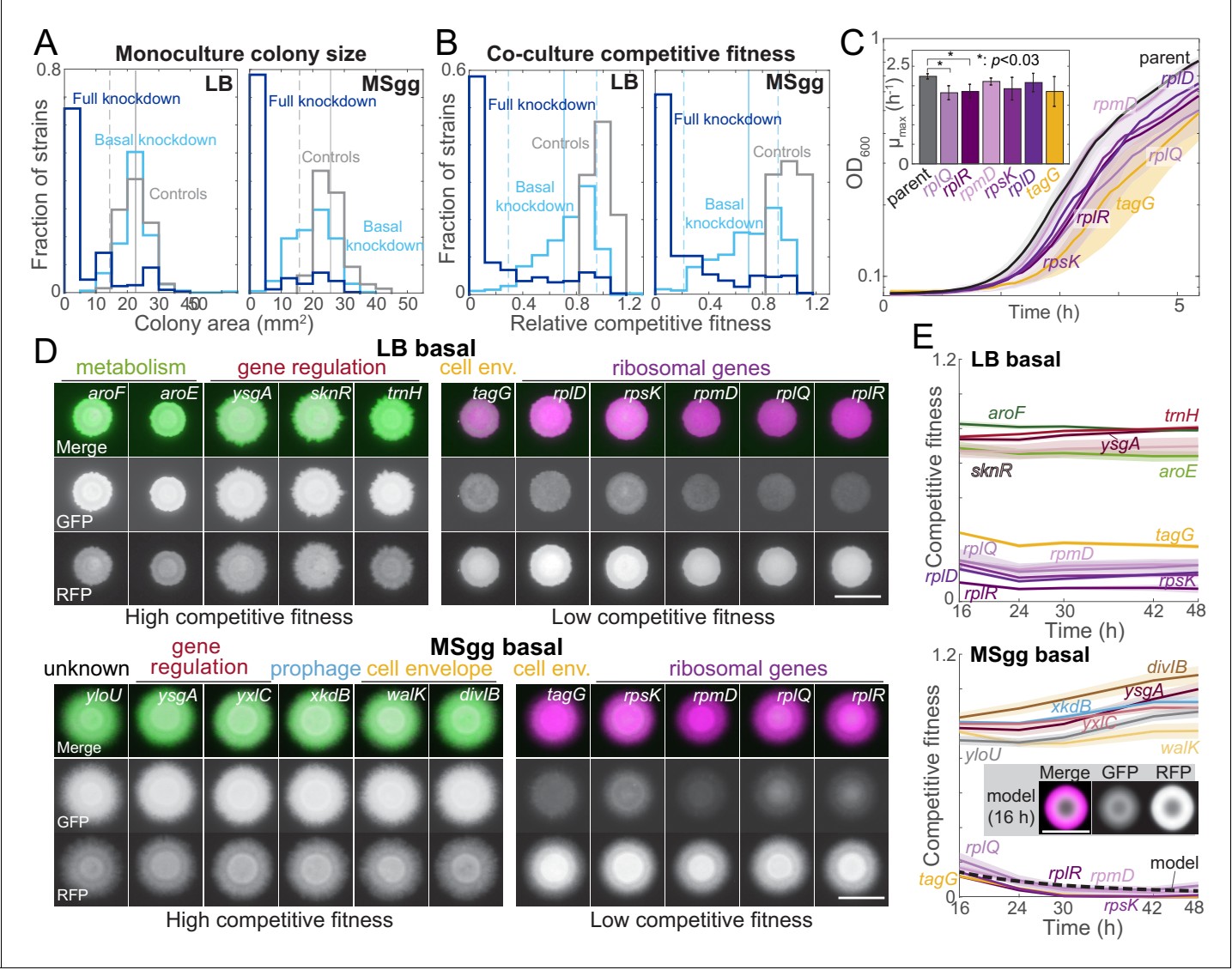

**Figure 2.** Growth on biofilm-promoting medium, increased knockdown, and competition against parent-RFP all broaden the distribution of fitnesses across the library. (A) Basal knockdown (light blue) of essential genes on LB agar (which does not promote biofilms) resulted in similar colony sizes as parent-GFP controls (gray); only 13 of 302 colonies had a colony size two standard deviations below the mean of the controls. By contrast, on biofilm-promoting MSgg agar the distribution of colony sizes spread to smaller values, with 80 colonies more than two standard deviations below the mean of the controls. Full knockdown (dark blue) inhibited growth of most strains. Data are from measurements at 16 hr using the *sacA::gfp* library. Vertical solid lines are means of the control distributions, and dashed lines show two standard deviations below the mean. (B) 17 (LB) and 11 (MSgg) knockdown strains in the library competed poorly against the parent-RFP strain at basal knockdown (light blue), while 41 (LB) and 46 (MSgg) had competitive fitness similar to parent-GFP+parent-RFP controls (gray) even at full knockdown (dark blue). Data are from competition ratios at 16 hr using the *sacA::gfp* library. Low-fitness strains were defined as having fitness at least two standard deviations below the mean of the data, and neutral-fitness strains were defined as having fitness at least one standard deviation above the mean. Vertical solid lines are means of the distribution from basal knockdown, and dashed lines show two standard deviations below and one standard deviation above the mean. (C) Strains with low competitive fitness for basal knockdown generally had lower growth rates in liquid monoculture than parent control strains. Colonies were inoculated into liquid LB, and $OD_{600}$ was monitored over time. Ribosome-related genes are shown in shades of purple, and a cell envelope-related gene (*tagG*) is shown in yellow. Curves are means, and shaded regions represent one standard deviation (n = 3). Inset: maximum growth rates. *: $p<0.03$, Student's unpaired *t*-test with a Benjamini–Hochberg multiple-hypothesis correction. (D) On both LB and MSgg agar, basal knockdown of *ygsA*, which is involved in gene regulation, exhibited high competitive fitness (left) and *tagG* and ribosomal-gene knockdowns exhibited low competitive fitness (right). GFP (knockdown strain) is false-colored green, and RFP (parent) is false-colored magenta. Images are from 16 hr using the *sacA::gfp* library. Scale bar: 5 mm. (E) Competitive fitness of the strains with the highest and lowest values was approximately constant after 16 hr. Curves are means, and shaded regions represent one standard error of the mean (n = 3 independent measurements). Inset and dashed black line: a reaction-diffusion model of co-culture colony growth with physically realistic parameters indicates that knockdowns (magenta) with maximum growth rate 20% lower than the parent (green) reproduce the colony

*Figure 2 continued on next page*

*Figure 2 continued*

size (bottom, inset) and competitive fitness (bottom, dashed black line) of ribosomal protein knockdowns after 16 hr (Materials and methods, *Figure 2—figure supplement 1C*).

The online version of this article includes the following figure supplement(s) for figure 2:

**Figure supplement 1.** Monoculture colony size screen, and analysis of knockdowns with low and high competitive fitness.

## Basal knockdown results in low competitive fitness for some depletions

Several strains competed poorly with the parent-RFP strain even under basal knockdown in both non-biofilm and biofilm colonies (*Figure 2B*). Interestingly, some non-essential genes had low competitive fitness. For instance, *mapA*, which encodes a methionine aminopeptidase, competed poorly in both LB and MSgg colonies, and CRISPRi induction further reduced fitness (*Supplementary file 2*, *Figure 2—figure supplement 1B*). Analysis of DAVID functional annotations of strains with competitive fitness >2 standard deviations below the mean of controls revealed significant enrichment of structural constituents of ribosomes ($p=9.8 \times 10^{-4}$ and $p=2.1 \times 10^{-2}$ on LB and MSgg agar, respectively). Some of the ribosomal protein knockdown strains that competed most poorly exhibited ~20% lower maximum growth rate than wild type in liquid cultures (*Figure 2C*), suggesting that the reduced competitive fitness of these strains is due to their reduced growth rate. Indeed, a reaction-diffusion model of colony growth of a co-culture indicated that a strain's maximum growth rate is a major determinant of competitive fitness, and that the 20% decrease of maximum growth rate in certain ribosomal protein knockdowns is consistent with our experimental measurements of their competitive fitness (*Figure 2E*, *Figure 2—figure supplement 1C*, Materials and methods).

## Some strains with reduced monoculture colony size do not have a fitness defect in co-culture

Several strains had fitness in co-culture with the parent-RFP strain similar to controls for basal and/or full knockdown (*Figure 2B*), suggesting that the targeted gene was rendered less essential by co-culture with parent-RFP. DAVID functional enrichment analysis of basal knockdowns with fitness values within one standard deviation of the mean across controls ($n$ = 41 and 46 strains for LB and MSgg, respectively) highlighted integral membrane components on both solid media ($p=2.1 \times 10^{-3}$ for both LB and MSgg). Under basal knockdown conditions, there were four strains (*menH* and *cytC* on LB and *aroF* and *rny* on MSgg) in which monoculture growth was clearly compromised by gene knockdown but competitive fitness remained high (*Figure 2—figure supplement 1B*). As *menH* and *aroF* are involved in the synthesis of menaquinone (vitamin K2) and aromatic amino acids, respectively, the high competitive fitness may result from nutrient sharing within the colony. In addition, aromatic amino acid biosynthesis genes were enriched in basal knockdowns with high competitive fitness on LB agar ($p=1.6 \times 10^{-2}$), potentially due to the presence of aromatic amino acids in rich LB medium but not in MSgg. These data underscore the medium dependence of gene essentiality in co-culture colonies.

## Competitive fitness is a reproducible and stable metric

To validate our findings, we replicated fitness measurements over time on a subset of strains with the highest or lowest competitive fitness values during basal knockdown on LB or MSgg agar. We found that the competitive fitness phenotype was highly reproducible and relatively stable over 2 days of colony growth (*Figure 2D, E*, *Figure 2—figure supplement 1D, E*), highlighting the utility of our CRISPRi library for probing the fitness of essential gene knockdown in co-cultures.

## Several biosynthesis-related genes have different phenotypes in rich versus minimal media

The distinct nutrient compositions of LB and MSgg, along with the much broader distribution of monoculture colony sizes in MSgg compared with LB (*Figure 2A*), motivated a comparison of competitive fitness across media. Somewhat surprisingly given the likelihood of different metabolic profiles due to media compositional differences, 93% of the strains exhibited similar competitive fitness (defined here as within 0.24 or 0.3 of the $y=x$ line for basal or full knockdown, respectively) on MSgg and LB agar, whether the targeted gene was basally or fully knocked down (*Figure 3A*,

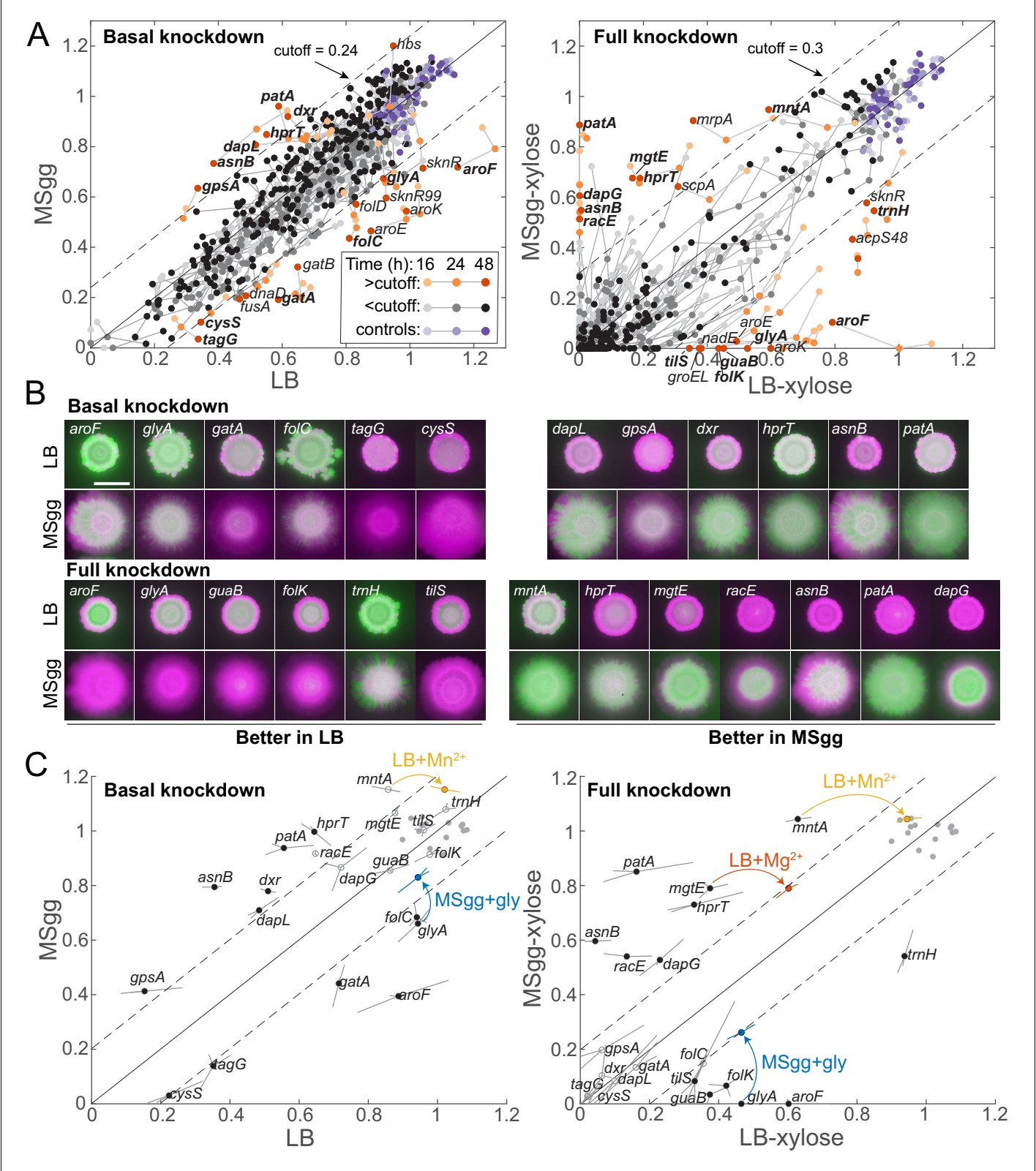

**Figure 3.** Some media-specific differences in competitive fitness can be directly attributed to the nutrient composition of the media. (**A**) Although most knockdowns had similar fitness on LB and MSgg agar, a subset of knockdowns had higher competitive fitness on LB agar than on MSgg agar, or vice versa. Genes with fitness difference >0.24 at 24 hr or >0.3 at 48 hr are annotated and colored in orange shades, while those below the cutoff are in grayscale and parent-RFP+parent-GFP co-culture controls are in shades of purple. Genes labeled in bold were selected for follow-up studies. Data are

*Figure 3 continued on next page*

*Figure 3 continued*

from the *sacA::gfp* library at 16, 24, and 48 hr. The solid line is y=x, and the dotted lines represent the chosen cutoff. (B) Images of colonies of the bolded genes in A after 48 hr that illustrate the differential competitive fitness between LB and MSgg. Green and magenta represent fluorescence from the gene knockdown and parent, respectively. Scale bar: 5 mm. (C) Addition of specific nutrients to the medium with lower competitive fitness rescued fitness for the *mntA, glyA,* and *mgtE* knockdowns. Means (circles) of triplicates (shown at the end of lines extending from the circle) are displayed. Filled black circles represent gene knockdowns that had substantially different competitive fitness in LB and MSgg. Open gray circles represent other genes tested that were not identified as different in that condition. Parent-RFP+parent-GFP controls are shown as gray filled circles. Data from addition experiments (LB+manganese, LB+magnesium, and MSgg+glycine for *mntA, mgtE,* and *glyA,* respectively) are shown as colored circles and lines at the ends of arrows. All changes marked with arrows are significant after correcting for multiple hypotheses with the Benjamini–Hochberg method (p<0.01, Student's unpaired *t*-test). Genes without a colored circle and arrow indicate that the fitness of that knockdown did not change upon addition of any of the nutrients tested (*Figure 3—figure supplement 1*).

The online version of this article includes the following figure supplement(s) for figure 3:

**Figure supplement 1.** Many nutrients do not impact the competitive fitness of knockdowns with different phenotypes between MSgg agar and LB agar.

*Supplementary file 2*). Nonetheless, we identified 36 strains that competed with the parent-RFP strain better on MSgg than on LB, or vice versa, under basal or full knockdown (*Figure 3A, B*, *Supplementary file 3*). Strains that competed better in one medium compared to the other were statistically enriched for genes involved in amino acid biosynthesis ($p=4.7 \times 10^{-6}$ and $p=8.3 \times 10^{-5}$ for basal and full knockdown, respectively), suggesting that some strains benefit from nutritional components specific to one medium (*Figure 3A*, *Supplementary file 3*).

Despite the undefined nature of LB, it was still possible in the case of many strains to identify candidate components whose addition to the medium with lower competitive fitness might rescue the deficit. We selected 20 knockdowns with medium-dependent fitness to pursue further, many of which displayed fitness differences for both basal and full knockdown (*Supplementary file 3*). The *glyA* knockdown had higher competitive fitness on LB agar than on MSgg agar, and as hypothesized, addition of glycine to MSgg significantly improved fitness in basal and induced conditions, to levels close to fitness on LB (*Figure 3C*). Similarly, adding $Mg^{2+}$ or $Mn^{2+}$ to LB agar at the same levels as in MSgg restored the competitive fitness of *mgtE* and *mntA* full knockdowns, respectively, to levels on MSgg (*Figure 3C*, *Figure 3—figure supplement 1A*, *Supplementary file 4*). While adding $Mn^{2+}$ to LB agar promotes biofilm phenotypes (*Shemesh and Chai, 2013*), in this case the additional $Mn^{2+}$ likely compensates for *mntA* knockdown as *mntA* is a component of the $Mn^{2+}$ ABC transporter (MntABCD) (*Que and Helmann, 2000*). Somewhat surprisingly, even though many of the remaining 17 knockdowns with medium-dependent fitness naturally suggested candidates for a missing nutrient, they did not exhibit increased fitness when the hypothesized nutrient was added to the medium with reduced fitness (*Figure 3—figure supplement 1B*, *Supplementary file 3,*), showing that at least exogenous provision of those nutrients is insufficient to complement the medium-specific fitness defect. Together, these results indicate that medium-dependent competition ratios can arise due to both nutrient compositional differences between media and other mechanisms that remain unidentified but highlight potentially important factors in selection during colony growth.

## Wild-type cells rescue *alrA* knockdown in a biofilm colony by sharing D-alanine

Since growth in a structured community provides opportunities for nutrient sharing and cellular differentiation, we hypothesized that some essential gene knockdowns would be unable to grow as a colony in monoculture but would fare better in co-culture with wild-type-like parent-RFP cells. Across our *sacA::gfp* essential gene knockdown library, we did not identify any knockdowns that exhibited robust growth in a colony co-culture but died as a monoculture (*Figure 1—figure supplement 2*, *Figure 2—figure supplement 1*, *Supplementary file 2*). (To conform to our inoculation protocol for biofilm colony cultures in which we used 1 µL of a liquid culture of OD ~1.0 [Materials and methods], our high-throughput screen involved inoculation of each ~2-mm-diameter spot with ~2 × 10^5 cells, likely facilitating the partial growth of some knockdowns that would be hampered in growth from a single cell on plates with xylose.) We found that disruption of the *thrC* locus prevents wrinkling formation on MSgg agar, presumably reflecting a growth defect, so we hypothesized that insertion of *gfp* at the *thrC* locus might exacerbate growth inhibition due to knockdown of certain essential

genes. Thus, we constructed a second *thrC::gfp* library of knockdowns of all 302 strains and screened it on MSgg with xylose.

In the *thrC::gfp* library, *alrA* was the only knockdown that failed to form a colony as a monoculture but survived with the parent-RFP strain in biofilm colonies on MSgg agar with xylose (*Figure 4A*, *Figure 4—figure supplement 1A, B*). AlrA is a racemase that converts L-alanine to D-alanine and is required for cross-linking of the peptidoglycan cell wall (*Figure 4B*). Full knockdown of *alrA* expression during liquid growth in a strain without *gfp* disruption of *thrC* led to bulging indicative of cell wall defects (*Figure 4C*, *Figure 4—figure supplement 1C*). On LB-xylose agar, the *alrA* strain from the *thrC::gfp* library managed to grow as a monoculture into a colony similar in size to the inoculation spot with petal-like outward projections (*Figure 4—figure supplement 1D*), as did

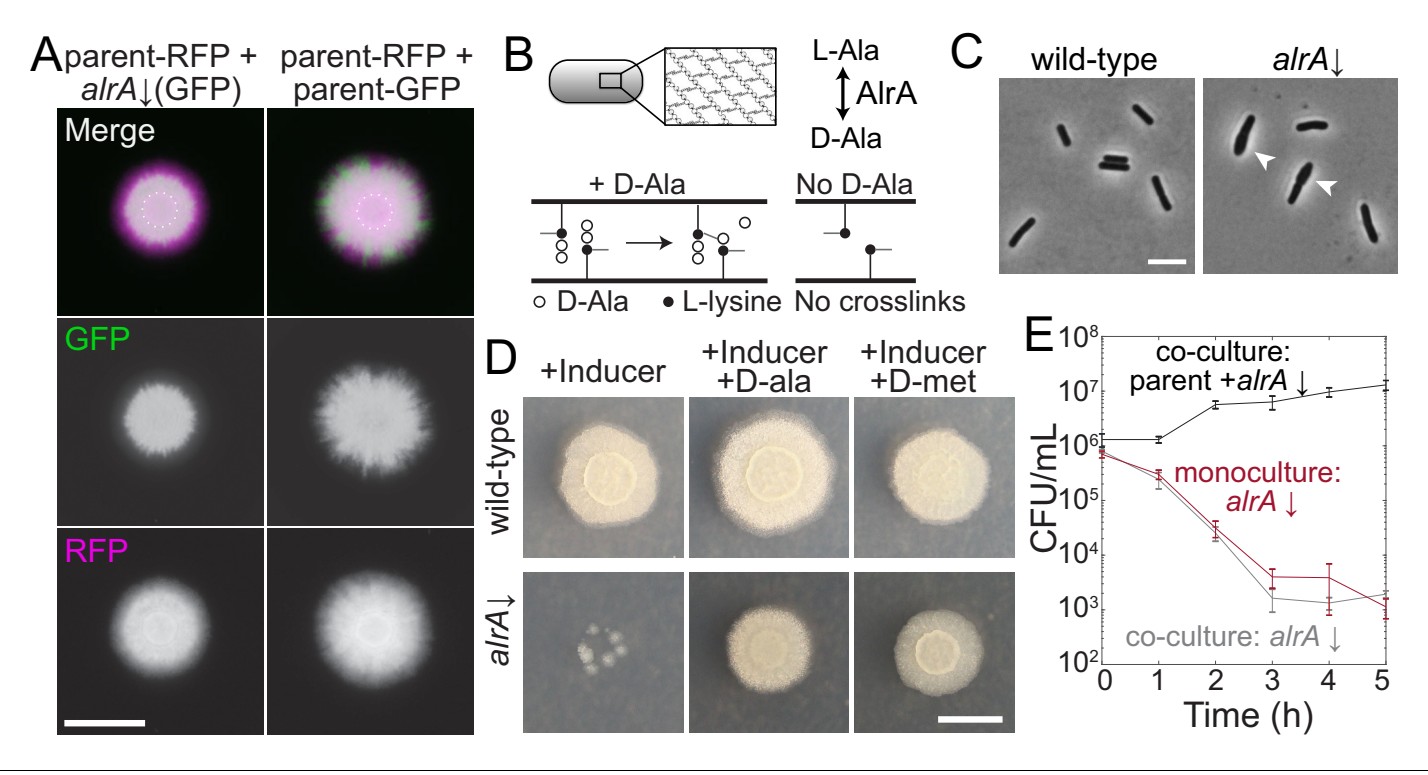

**Figure 4.** Full knockdown of *alrA* is rescued by D-alanine nutrient sharing in a biofilm colony, but not in liquid culture. (**A**) Left: the *sacA::gfp alrA* knockdown under full knockdown was rescued by growth with the parent-RFP strain under biofilm-promoting conditions (MSgg-xylose agar). The *alrA* knockdown expanded beyond the boundaries of the original inoculum (dashed circle) when grown in co-culture with the parent-RFP strain. Right: the control co-culture of parent-RFP with parent-GFP preserved both strains at approximately equal proportions. Images were acquired at 24 hr. In merged images, GFP from the *alrA* knockdown is false-colored green and RFP from the parent-RFP strain is false-colored magenta. Scale bar: 5 mm. (**B**) AlrA is a racemase that converts L-alanine to D-alanine. D-alanine is critical for cell wall cross-linking. (**C**) Full knockdown of *alrA* caused cells to bulge, signifying cell wall defects. Cells were cultured for 6 hr in liquid MSgg with xylose to fully inhibit *alrA* expression. Arrowheads indicate bulging cells. Scale bar: 5 μm. (**D**) Full knockdown of *alrA* was rescued by exogenous D-alanine. Cultures were grown in liquid LB to an $OD_{600}$ ~1 and then 1 μL was spotted on MSgg xylose agar alone or supplemented with 0.04 mg/mL D-alanine or D-methionine. Cells from *alrA* monocultures mostly died (left); the small colonies represent suppressors present in the initial inoculum. By contrast, addition of D-alanine (middle) or D-methionine (right) resulted in comparable growth to wild type. Images are of an unlabeled *alrA* knockdown (HA420) and were acquired after 24 hr of growth. Scale bar: 5 mm. (**E**) Full knockdown of *alrA* was not rescued when co-cultured with the parent-RFP strain in liquid. For the co-culture, parent and *alrA* knockdown cultures were mixed 1:1 and back-diluted 1:100 into liquid MSgg with xylose to fully deplete *alrA*. For the *alrA* knockdown monoculture, the culture was diluted 1:200 into liquid MSgg with xylose so that the starting inoculum of the *alrA* strain was equivalent to that of the co-culture. CFU/mL of the *alrA* knockdown were not significantly different between the monoculture (dark red) and co-culture (gray) throughout the course of the experiment (*p*-values from each time point range from 0.21 to 0.66, student's unpaired *t*-test). The black line is the total CFU/mL of the parent/*alrA* knockdown co-culture. *n* = 3, error bars represent one standard error of the mean.

The online version of this article includes the following figure supplement(s) for figure 4:

**Figure supplement 1.** Under full knockdown, the *alrA* knockdown dies as a monoculture colony but grows when co-cultured with the wild-type-like parent.

the *alrA* strain from the *sacA::gfp* library on LB-xylose and MSgg-xylose plates (*Figure 2—figure supplement 1A*). As discussed above, the absence of complete lysis is likely due to the high initial density driving growth of a visible colony (*Figure 2—figure supplement 1A*); streaking all *alrA* strains to reduce initial cell density resulted in a substantial reduction in the number of colonies on plates with xylose compared to without for both LB and MSgg (*Figure 4—figure supplement 1E*), indicating that the full-knockdown phenotype across all media and genotypes is severe for *alrA* at lower initial cell densities.

We hypothesized that cells with full knockdown of *alrA* transcription were able to maintain their growth in biofilm colony co-culture because the parent-RFP cells were providing the necessary D-alanine. To test this hypothesis, we grew monocultures of an *alrA* strain without *gfp* in the genome on MSgg-xylose with exogenous D-alanine and found that D-alanine rescued biofilm colony growth (*Figure 4D*). D-methionine, an amino acid that can substitute for D-alanine in cell wall cross-linking (*Cava et al., 2011*), also rescued *alrA* growth on MSgg-xylose plates (*Figure 4D*), while other D-amino acids that are not incorporated into the cell wall did not rescue colony growth (*Figure 4—figure supplement 1F, G*), suggesting that D-alanine's specific role in peptidoglycan synthesis is rescued. Thus, sharing of D-alanine within a biofilm colony rescues *alrA*-depleted cells, likely by stabilizing mutant cell walls.

To test whether *alrA* cells are rescued by the parent-RFP strain in liquid MSgg-xylose, we grew liquid co-cultures and plated dilutions at hourly time points to quantify survival (separating the two strains based on fluorescence). The vast majority of *alrA* cells died within hours in both liquid monocultures and co-cultures with the parent-RFP strain (*Figure 4E*). Thus, the rescue of *alrA* knockdown cells by D-alanine sharing in co-cultures is specific to growth in a colony, presumably due to the close proximity of cells that facilitates D-alanine sharing.

## *alrA* knockdown cells stably co-exist with extracellular matrix-deficient wild-type cells in a biofilm colony

Secretion of extracellular matrix provides structural integrity to biofilm colonies (*Flemming and Wingender, 2010*), including for *B. subtilis* strain 3610 on MSgg agar (*Srinivasan et al., 2018*). Thus, we wondered if matrix plays a role in the rescue of *alrA*, either by providing structural support to *alrA* knockdown cells with weaker walls (*Figure 4C*) or by facilitating the diffusion of D-alanine. To test this idea, we deleted the genes encoding both of the main extracellular matrix components (EpsH, TasA) from the parent-RFP strain and the *alrA* knockdown in the *sacA::gfp* wrinkling-proficient library. We mixed the two matrix-deficient strains and quantified colony size and competitive fitness in full knockdown conditions on MSgg-xylose plates. As expected, matrix-deficient co-culture colonies were smaller than matrix-proficient co-cultures as matrix is necessary for robust biofilm colony growth (*Seminara et al., 2012*; *Figure 5A*). Nonetheless, matrix-deficient co-cultures exhibited approximately the same fraction of *alrA* cells as matrix-proficient co-cultures (*Figure 5A, B*, *Figure 5—figure supplement 1A*). Thus, matrix is not required for the growth rescue of *alrA*-knockdown cells.

Surprisingly, combining the matrix-proficient and wrinkling-proficient *alrA* strain under full knockdown with the matrix-deficient (Δ*epsH* Δ*tasA*) parent-RFP strain resulted in an increased fraction of *alrA*-depleted cells relative to co-cultures with the matrix-proficient parent-RFP strain. By contrast to when both strains were matrix-proficient and the parent-RFP strain outcompeted *alrA*-depleted cells at the colony edge, *alrA*-depleted cells were distributed throughout the entire colony (*Figure 5A*). In addition to improved growth of the *alrA* knockdown, the RFP fluorescence of the matrix-deficient parent-RFP was also higher and occupied a larger area than in the co-culture of matrix-deficient *alrA* and parent-RFP strains, suggesting that the matrix-capable mutant promotes growth of the matrix-deficient parent strain (*Figure 5A*). Moreover, rather than a steady decrease in competitive fitness over time (*Figure 5A,B*), the competitive fitness of *alrA* against the matrix-deficient parent was stable at ~1 for 48 hr. The increased and stable fitness of this strain combination as a biofilm colony suggests a synthetic mutualistic interaction, defined here as the fitness of both strains in co-culture being stable and higher than either strain on its own. The matrix-deficient parent-RFP cells restore viability to *alrA* knockdown cells by providing D-alanine, and in turn *alrA* knockdown cells enhance growth of the parent-RFP strain by providing extracellular matrix.

Our finding that the competitive fitness of *alrA* decreased over time in a co-culture with parent-RFP in which both strains were matrix-deficient (and hence did not form a canonical biofim colony)

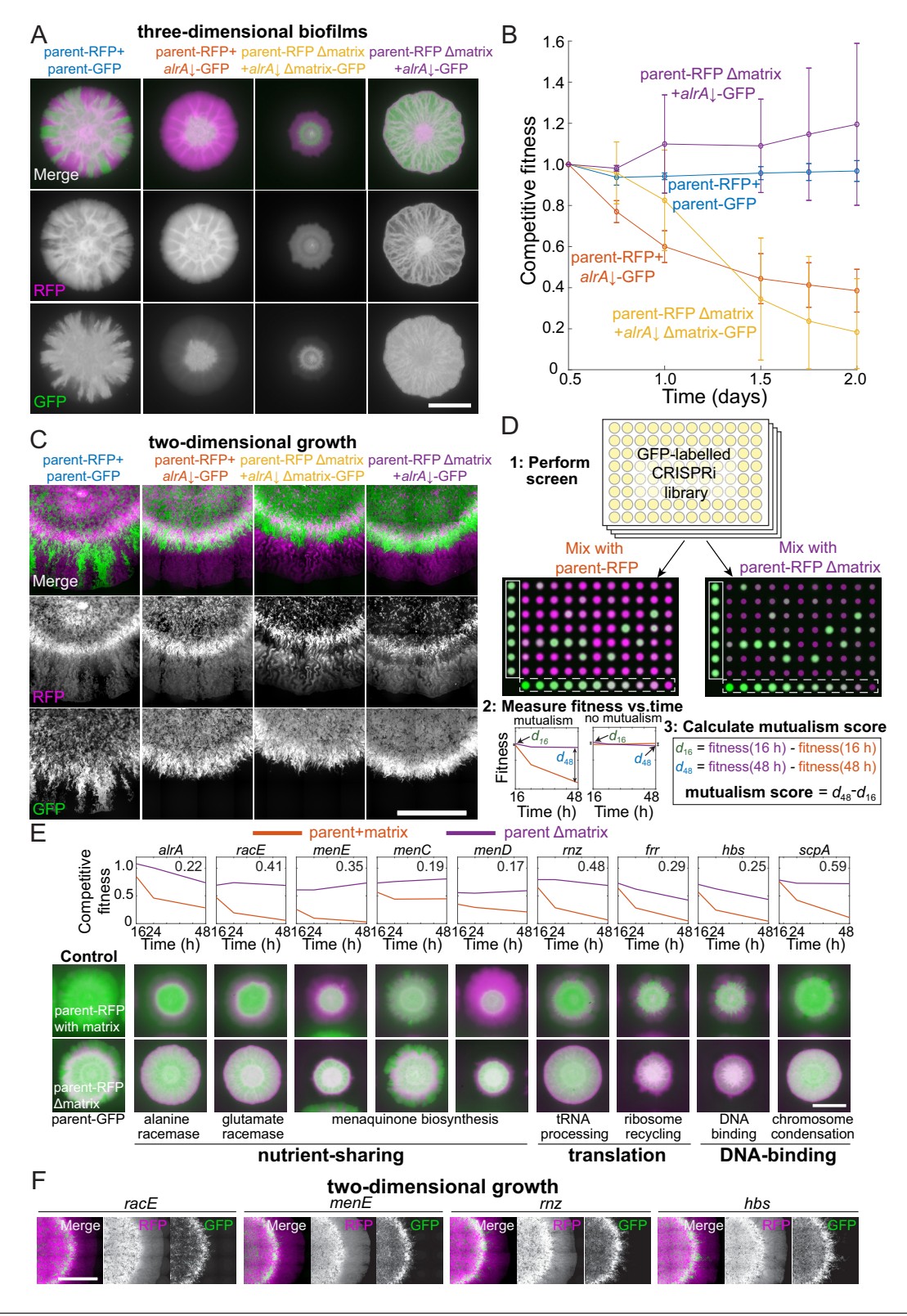

**Figure 5.** Mutualisms emerge when a nutrient-deficient mutant is the sole provider of extracellular matrix. (**A**) Extracellular matrix is not required for *alrA* rescue (third column), and rescue is enhanced when the *alrA* full knockdown is combined with a matrix-deficient parent strain (fourth column). In the left column, the parent-GFP strain is false-colored in green, and in the other columns the *alrA* knockdown strain is green. The parent-RFP strain is false-colored in magenta. In the first two columns, both strains express matrix proteins. In the third column, both strains lack *epsH* and *tasA*, which

*Figure 5 continued on next page*

*Figure 5 continued*

encode key matrix components. In the fourth column, *epsH* and *tasA* are deleted from the parent-RFP strain while the *alrA* knockdown produces matrix. Images were acquired after 48 hr of growth. Scale bar: 5 mm. (B) Competitive fitness of the *alrA* full knockdown decreased over time when both strains or neither produced matrix, while fitness remained 1 (equal proportions of the two strains) and stable over time in co-cultures when the parent was matrix-deficient. The matrix-deficient parent-RFP+*alrA* knockdown (purple) and the parent-RFP+parent-GFP control (blue) data were not significantly different ($p>0.2$ at all time points); nor were the parent-RFP+*alrA* knockdown (orange) and the matrix-deficient parent-RFP+matrix-deficient *alrA* knockdown (yellow) data ($p>0.07$, Student's *t*-test). The matrix-deficient parent-RFP+*alrA* knockdown (purple) fitness data were significantly higher than the parent-RFP+*alrA* knockdown (orange) at all time points after 0.5 days ($p<0.014$). Data were normalized to fitness at 0.5 days. Curves are means, and error bars represent one standard deviation ($n$ = 2–3 biological replicates). Statistical analysis: Student's unpaired *t*-test with a Benjamini–Hochberg multiple-hypothesis correction. (C) Cells with full knockdown of *alrA* were outcompeted by the parent-RFP strain during two-dimensional growth. Co-cultures were spotted on a MSgg-xylose agar pad and limited to growth in a layer with thickness one cell by applying a cover slip over the cells. Images were acquired after 24 hr of growth. Scale bar: 0.4 mm. (D) Design of screen to identify mutants that exhibit an increase in fitness when co-cultured with a matrix-deficient parent on MSgg-xylose agar. The *sacA::gfp* library was used in this screen; plate 3 from the 16 hr time point is shown. The distance between the centers of each colony is 9 mm. White box, controls; white dashed box, titration row. (E) Full knockdowns that exhibited mutualism generally involve genes related to nutrient sharing, translation, and DNA binding. Top: competitive fitness of co-cultures with the wild-type parent-RFP (purple) and matrix production-deficient parent (red) at 16, 24, and 48 hr. Numbers in the top right indicate the mutualism score. Bottom: merged images of biofilm colony co-cultures on MSgg-xylose agar at 48 hr. Scale bar: 5 mm. (F) Full knockdowns that exhibited mutualism in three-dimensional biofilm colonies were outcompeted by the matrix-deficient parent-RFP strain when growth was limited to two dimensions as in C. Images were acquired after 24 hr of growth. Scale bar: 0.4 mm. For (A, C–F), GFP is false-colored in green and RFP is false-colored in magenta.

The online version of this article includes the following figure supplement(s) for figure 5:

**Figure supplement 1.** A mutualism screen reveals full knockdowns with improved growth in co-culture when the parent is deficient in production of extracellular matrix.

(*Figure 5A*) suggests that the mutualism between *alrA* and the parent-RFP strain requires growth in a matrix-capable biofilm colony. A recent study showed that biofilm colony expansion in three dimensions depends heavily on extracellular matrix, while two-dimensional growth relies more on cell growth and division (*Srinivasan et al., 2019*). To test whether the mutualism between *alrA* and the matrix-deficient parent was dependent on three-dimensional biofilm colony growth, we grew co-cultures between an agar pad and a coverslip. In this configuration, the growing edge of the colony has a thickness of only one cell, essentially constraining growth to two dimensions (*Figure 5—figure supplement 1B*). Now, in all matrix-production combinations of the *sacA::gfp alrA* strain under full knockdown and parent-RFP, including matrix-proficient *alrA* with the matrix-deficient parent, the parent-RFP strain had a growth advantage at the edge of the two-dimensional colony and outcompeted the *alrA* knockdown (*Figure 5C, F*). Thus, the mutualism we discovered between *alrA* under full knockdown and the matrix-deficient parent can be eliminated by removing the ability of the colony to grow in three dimensions.

## Many metabolism-related mutants exhibit enhanced rescue in three-dimensional biofilm colonies with matrix-deficient parent cells

Our discovery that *alrA* knockdown cells grew in a mutualism with an extracellular matrix-deficient parent (*Figure 5A, B*) led us to hypothesize that other essential gene knockdowns might display enhanced growth when cultured with the matrix-deficient parent. To test this hypothesis, we performed co-culture screens of each strain in the *sacA::gfp* library with either the matrix-proficient or matrix-deficient parent-RFP strain on MSgg-xylose (*Figure 5D*, *Figure 5—figure supplement 1C*, *Supplementary file 2*). We measured competitive fitness at 16, 24, and 48 hr to identify strains that had increased and relatively stable fitness when they were the sole provider of extracellular matrix relative to competition with the wild-type-like parent-RFP. For each strain, we calculated a mutualism score defined as the fitness increase due to deletion of matrix components from the parent at 48 hr compared with the fitness increase at 16 hr; a positive score reflects a growing benefit of being the sole matrix provider, and hence implies relatively stable fitness (*Figure 5D*, *Figure 5—figure supplement 1D*). We focused on all strains with a mutualism score two standard deviations above the mean across controls (>0.22, *Figure 5—figure supplement 1D*).

On MSgg-xylose, in addition to *alrA*, eight other essential gene full knockdowns exhibited a high mutualism score (*Figure 5E*, *Figure 5—figure supplement 1E, F*, *Supplementary file 5*). Of these strains, knockdowns of genes encoding the glutamate racemase (*racE*) and enzymes involved in menaquinone (vitamin K2) synthesis (*menE*) were candidates for nutrient- and matrix-sharing

mutualisms similar to that of *alrA*. Knockdowns of two other genes in the menaquinone synthesis pathway (*menC* and *menD*) displayed mutualism scores slightly below 0.22 (*Figure 5E*, *Supplementary file 5*), providing further support for a menaquinone-based mutualism. The other essential gene knockdowns with high mutualism scores encode proteins that bind DNA (*scpA* and *hbs*) or are related to translation (*rnz* and *frr*); these genes likely either play an indirect role in regulating biosynthesis of a shared nutrient to support the mutualism or employ other mechanisms outside of nutrient sharing. Strains depleted of *racE*, *menE*, *menD*, and *rnz* showed evidence of being outcompeted by parent-RFP, but stably coexisted throughout the entire colony when the parent could not produce matrix (*Figure 5E*), further suggesting that matrix-capable mutants promote growth of the matrix-deficient parent strain. In addition, four non-essential strains exhibited high mutualism scores (*Figure 5—figure supplement 1F*), indicating that the benefits of matrix sharing can extend to genes that are not critical for growth as a monoculture.

Since the *alrA* full knockdown exhibited mutualism in a three-dimensional biofilm colony but not when growth was confined to two dimensions, we tested whether the other mutualisms were maintained during two-dimensional growth. We grew *racE*, *menE, rnz,* and *hbs* full depletions in individual co-cultures with the matrix-deficient parent-RFP between a glass slide and a coverslip. In each case, the knockdown was outcompeted by the parent-RFP strain at the growing edge of the colony (*Figure 5F*). In sum, these data suggest that matrix-dependent mutualisms generally require growth in a three-dimensional colony.

## Discussion

Here, we created two libraries of essential gene knockdowns in a biofilm-capable *B. subtilis* strain and developed a powerful high-throughput screen of competition in bacterial co-cultures to reveal genetic interactions specific to growth in three-dimensional colonies. First, we showed that basal knockdown of some ribosomal proteins reduces competitive fitness with a wild-type-like strain in co-culture colonies (*Figure 2*), suggesting a high degree of selection on these genes during colony growth. Second, we found that medium composition can dramatically alter competition (*Figure 3*), highlighting the role of the extracellular environment on fitness in a multicellular context. Third, we discovered that knockdown of *alrA* can be rescued through sharing of D-alanine in a three-dimensional colony, a context in which the gene is 'less essential,' but not in liquid or when growth is confined to two dimensions between an agar surface and a coverslip (*Figures 4* and *5*). Finally, we uncovered a mutualism between *alrA* knockdown cells and a parent deficient in extracellular matrix production based on sharing of nutrients and matrix components, and used this finding to identify several other essential gene knockdowns exhibiting similar mutualistic interactions (*Figure 5*). These findings illustrate how growth in a biofilm colony can alter natural selection by supporting mutant cells that are less likely to survive on their own through short-range interactions.

Despite previous studies showing that D-alanine levels are undetectable in *B. subtilis* 168 liquid culture supernatants (*Lam et al., 2009*), we found that D-alanine sharing in a colony can rescue *B. subtilis* 3610 mutants that cannot synthesize their own D-alanine. Thus, D-alanine is produced and secreted by wild-type cells at sufficient levels to support growth of the mutant in a colony on biofilm-promoting media. Since full knockdown of *alrA* causes cells to bulge and die in liquid culture within hours (*Figure 4C, E*), we infer that rescue likely occurs early in biofilm development prior to the period of substantial cell death that is thought to drive wrinkling (*Asally et al., 2012*), suggesting that rescue is not due to the release of D-alanine by dying cells. The close proximity of cells within the colony may aid in rescue, even if secreted D-alanine levels are low. Regardless, this rescue demonstrates that cell wall synthesis mutants can be supported in native environments through sharing of cell wall components, which could be provided by many other bacterial species in a multispecies community due to the common chemical makeup of peptidoglycan cell walls (*Cava et al., 2011*).

Our observation that rescue of *alrA*-depleted cells did not occur when growth was constrained to two dimensions (*Figure 5C*), combined with the finding that rescue occurred in co-culture colonies when both the *alrA* knockdown and the parent were matrix-deficient (*Figure 5A, B*), indicates that some aspect of three-dimensional growth beyond matrix production is fundamental to the rescue. One possibility is liquid uptake facilitated by colony architecture, which has been hypothesized to act like a sponge and thereby drive colony expansion (even more so when extracellular matrix is present) (*Seminara et al., 2012*; *Yan et al., 2017*). It is also possible that cellular differentiation and

development are disrupted by limiting growth to a thin layer. Irrespective, our findings highlight the importance of future high-throughput genetic screens that embrace the natural context of three-dimensional colony growth on surfaces.

The stable mutualism that we discovered between the *alrA* knockdown and a matrix-deficient parent (*Figure 5B*) resembles the initial behavior of Δ*tasA* and Δ*epsH* mutants grown together as pellicle biofilms on liquid surfaces, in which the pellicle architecture is preserved by cross-complementation for many passages (*Dragoš et al., 2018*). Our discovery of matrix-based mutualisms involving multiple genes with a range of cellular functions (*Figure 5E*) motivates future studies to probe the nature of the *tasA-espH* interaction in these interactions, specifically to examine which matrix components are most important and whether mutualisms can be sustained through repeated passages or with different starting ratios of the strains in the co-culture. We expect the genes involved in nutrient-sharing mechanisms (e.g., with D-alanine and menaquinone) to stably interact in passaging experiments as cross-complementation is important for the fitness of both strains. Importantly, our observation that these mutualisms appear to generally require growth in three dimensions further highlights the importance of the geometry of the native environment during evolution.

Together, our results demonstrate that growth in a biofilm colony can drive genetic diversity and illustrate the potential for mutualism between nutrient and matrix sharing in native biofilms. Such mutualisms may occur during plant root colonization, when the bacterial extracellular matrix is particularly important and may serve to pull nutrients from the root and surrounding soil (*Beauregard et al., 2013*). In addition to the potential implications for plant growth-promoting bacteria in the rhizosphere, this study provides a foundation to understand how microbial biofilm growth affects selection in industrial and clinical settings.

# Materials and methods

## Key resources table

| Reagent type (species) or resource | Designation | Source or reference | Identifiers | Additional information |
|---|---|---|---|---|
| Strain, strain background (*Bacillus subtilis*) | *Bacillus subtilis* strain NCIB 3610 | Daniel B. Kearns laboratory | NCIB 3610 or 3610 | *trpC+*, *rapP+*, *sfp+*, *epsC+*, *swrA+*, *degQ+*, pBS32 |
| Strain, strain background (*Bacillus subtilis*) | Parent-RFP | This work | HA12 | Genotypes listed in table S1 |
| Strain, strain background (*Bacillus subtilis*) | Parent-GFP (*sacA*) | This work | HA773 | Genotypes listed in table S1 |
| Strain, strain background (*Bacillus subtilis*) | *sacA::gfp* CRISPRi library | This work | See table S1 | Genotypes listed in table S1 |
| Strain, strain background (*Bacillus subtilis*) | Parent-GFP (*thrC*) | This work | HA49 | Genotypes listed in table S1 |
| Strain, strain background (*Bacillus subtilis*) | *thrC::gfp* CRISPRi library | This work | See table S1 | Genotypes listed in table S1 |
| Strain, strain background (*Bacillus subtilis*) | *rfp* CRISPRi knockdown | This work | HA13 | Genotypes listed in table S1 |
| Strain, strain background (*Bacillus subtilis*) | *alrA* CRISPRi knockdown | This work | HA420 | Genotypes listed in table S1 |
| Strain, strain background (*Bacillus subtilis*) | *alrA* CRISPRi knockdown Δ*matrix* | This work | HA827 | Genotypes listed in table S1 |

*Continued on next page*

*Continued*

| Reagent type (species) or resource | Designation | Source or reference | Identifiers | Additional information |
|---|---|---|---|---|
| Strain, strain background (*Bacillus subtilis*) | Parent-RFP Δ*matrix* | This work | HA825 | Genotypes listed in table S1 |
| Genetic reagent (SPP1 phage) | SPP1 phage | Daniel B. Kearns laboratory | | |
| Recombinant DNA reagent | pDG1731-gfp | This work | | $P_{veg}$-*sfGFP* in a *thrC* integration construct |
| Recombinant DNA reagent | pDG1731 | Bacillus genetic stock center | | |
| Sequence-based reagent | sfGFP fw | This work | | tcctagaagcttatcgaattc CTTATTAACGTTGATATAATTT AAATTTTATTTGACAAAAATGG GCTCGTGTTGTACAATAAATGT AACTACTAGTACATAAGGAGGAA CTACTATGAGCAAAGGAGAAGAACTTTTC |
| Sequence-based reagent | sfGFP rev | This work | | ttaagcaccggtttatta TTTGTAGAGCTCATCCATGCC |
| Chemical compound, drug | LB lennox medium | RPI | L24066-1000.0 | |
| Software, algorithm | Matlab | MathWorks | R2018a | Some scripts were written in earlier versions of Matlab, but all are compatible with R2018a |
| Software, algorithm | Adobe Photoshop | Adobe, Inc | CS6 | |
| Software, algorithm | FIJI | ImageJ | 2.0.0-rc-44/1.50e | |
| Software, algorithm | DAVID functional annotation tool | https://david.ncifcrf.gov | | |
| Software, algorithm | Custom scripts for growth and image analysis | This work | | Data Dryad: doi:10.5061/dryad.79cnp5htm |

## Media

Strains were grown in LB (Lennox lysogeny broth with 10 g/L tryptone, 5 g/L NaCl, and 5 g/L yeast extract) or a defined MSgg medium (5 mM potassium phosphate buffer, diluted from 0.5 M stock with 2.72 g $K_2HPO_4$ and 1.275 g $KH_2PO_4$, pH 7.0 in 50 mL; 100 mM MOPS buffer, pH 7.0, adjusted with NaOH [note that some laboratories adjust the pH of MOPS buffer with KOH]; 2 mM $MgCl_2 \cdot 6H_2O$; 700 µM $CaCl_2 \cdot 2H_2O$; 100 µM $FeCl_3 \cdot 6H_2O$; 50 µM $MnCl_2 \cdot 4H_2O$; 1 µM $ZnCl_2$; 2 µM thiamine HCl; 0.5% [v/v] glycerol; and 0.5% [w/v] monosodium glutamate). MSgg medium was made fresh from stocks the day of each experiment for liquid cultures or a day before the experiment for agar plates. Glutamate and $FeCl_3$ stocks were made fresh weekly. Colonies were grown on 1.5% agar plates. For nutrient addition assays (*Figure 3*, *Figure 3—figure supplement 1*), we supplemented LB with one of the following: 0.5% (w/v) monosodium glutamate, 2 mM $MgCl_2 \cdot 6H_2O$, 50 µM $MnCl_2 \cdot 4H_2O$, 2 mM $MgCl_2 \cdot 6H_2O$, 0.5% (w/v) L-asparagine, 0.5% (w/v) L-aspartic acid, 0.5% (w/v) L-lysine, or 0.5% (w/v) D-glutamic acid, and we supplemented MSgg with one of the following: 0.5% (w/v) L-cysteine, 0.5% (w/v) L-glutamine, 0.5% (w/v) L-glycine, 0.5% (w/v) L-serine, or 0.5% (w/v) L-tryptophan. Where indicated, L-threonine was added to MSgg at a concentration of 0.1 mg/mL. D-amino acids (D-alanine, D-methionine, D-glutamate, D-leucine, D-serine, D-valine) were each used at a concentration of 0.04 mg/mL. We made TY medium for phage transduction using the LB recipe above supplemented with 10 mM $MgSO_4$ and 0.1 mM $MnSO_4$.

Antibiotics for selection of mutant strains were used as follows: kanamycin (kan, 5 µg/mL), MLS (a combination of erythromycin at 0.5 µg/mL and lincomycin at 12.5 µg/mL), chloramphenicol (cm, 5 µg/mL), tetracycline (tet, 12.5 µg/mL), and spectinomycin (spc, 100 µg/mL).

## Strain construction

All strains and their genotypes are listed in *Supplementary file 1*. For library construction, we used SPP1 phage transduction (*Yasbin and Young, 1974*). We used a 168 -strain containing $P_{xyl}$-dCas9 at the *lacA* locus (CAG74399) as a donor and wild-type strain 3610 (a gift from Dan Kearns) as the recipient to create the 3610-dCas9 parent strain (CAG74331) using MLS for selection. We then used this 3610-dCas9 parent as the recipient and a strain with $P_{hyper-spank}$-gfp at the *sacA* locus (NRS1473, a gift from Nicola Stanley Wall) or a strain with $P_{veg}$-gfp at the *thrC* locus as the donor strain to create the parent-GFP strain expressing *dCas9* and *gfp* (HA773 is the *sacA::gfp* parent and HA47 is the *thrC::gfp* parent; phage transduction described below), using kanamycin and tetracycline for selection, respectively.

For construction of mutant strains, we used either the *sacA::gfp* parent or the *thrC::gfp* parent as the recipient strain and strains from a 168 CRISPRi library (*Peters et al., 2016*) as the donor. We amended the phage transduction protocol to increase the throughput of strain construction as follows. We grew donor strains in 96-well deep-well plates (1 mL cultures in TY medium in 2 mL wells) for at least 5 hr shaking at 37°C with a Breath-easy (Sigma-Aldrich, St. Louis, MO, USA) film covering the plate. We then aliquoted 0.1 mL of $10^{-5}$ dilutions of fresh phage stocks grown on strain 3610 cells ($10^{-5}$ was chosen as the dilution factor because it provided the appropriate level of lysis for our phage stock in a trial transduction) into 77 or 71 glass test tubes (each plate of the library contains 77 strains, except the fourth plate contains 71 strains). We added 0.2 mL of each culture to the tubes and incubated the entire rack at 37°C for 15 min. Then, working quickly in batches of 11, we added 4 mL of TY molten soft agar (~55°C) to each phage-cells mixture, mixed gently, and poured onto TY plates so that the soft agar covered the entire plate. We incubated these plates at 37°C overnight in a single layer (not stacked). The next day we examined the plates for lysis and added 5 mL TY broth with 250 ng DNase to each plate and scraped the top agar with a 1 mL filter tip to liberate phage. We then pipetted the TY broth into a syringe attached to a 0.45 µm filter and carefully filtered into a 5 mL conical vial. After filtering, 1 mL of lysate was added to the appropriate well of a deep-well 96-well plate. Once all of the phage was isolated, we arrayed 10 µL of each phage stock into 96-well microtiter plates. We aliquoted 100 µL of a saturated (>5 hr of culturing, $OD_{600}$ >1.5) culture into the wells containing phage and incubated for 25 min at 37°C without shaking. We plated the phage/cell mixtures onto selection plates (LB with chloramphenicol and citrate to select for the sgRNA locus) and incubated the plates for 18 hr at 37°C. Any plates that did not have visible colonies after this incubation were incubated further at room temperature, and colonies generally appeared within a day. We streaked transductant colonies for single colonies onto LB+chloramphenicol plates and stocked a single colony for each strain in the library by growing in 5 mL LB on a roller drum at 37°C to mid- to late-log phase and then adding the culture to the appropriate well of 96-well plate with a final concentration of 15% glycerol. The library was stocked at −80°C.

The GFP-labeled Δ*epsH* Δ*tasA alrA* knockdown strain (HA823) and the parent-RFP Δ*epsH* Δ*tasA* strain (HA825) were constructed using phage transduction as described above using DS9259 and DS3323 (gifts from Dan Kearns) as donor strains for *epsH::tet* and *tasA::tn10spc*, respectively, and the *sacA::gfp alrA* knockdown strain (HA761) or parent-RFP (HA12) as the recipient. The *epsH::tet* transduction was performed first, and the resulting strains were used as the parent to add the *tasA:: tn10spc* construct.

To construct plasmid pDG1731-gfp ($P_{veg}$-sfGFP in a *thrC* integration construct), the following primers were used to clone superfolder GFP (sfGFP) and add the $P_{veg}$ promoter: forward, tcctagaagct-tatcgaattcCTTATTAACGTTGATATAATTTAAATTTTATTTGACAAAAATGGGCTCGTGTTGTACAA TAAATGTAACTACTAGTACATAAGGAGGAACTACTATGAGCAAAGGAGAAGAACTTTTC; reverse, ttaagcaccggtttattaTTTGTAGAGCTCATCCATGCC. The amplicon and pDG1731 were both digested with HindIII and AgeI and ligated together. The ligation was used to transform chemically competent *Escherichia coli*. We transformed *B. subtilis* 168 with pDG1731-gfp to create HA45 and confirmed double crossover (spc$^R$, MLS$^S$), then used HA45 as the donor and HA2 as the recipient in phage transduction to create the *thrC* parent-GFP strain ($P_{xyl}$-dCas9 *thrC::$P_{veg}$-gfp*, HA47).

## Growth conditions for library screens of growth on agar plates

To grow the library for monocultures and co-cultures, we inserted a sterile 96-well Singer pin (Singer Instruments, Somerset, UK) into frozen glycerol stocks and applied pressure and agitation so that

each pin picked up some of the frozen glycerol stock from the appropriate well. The Singer pin was used to spot onto LB agar in a rectangular Singer plus plate, and the plate was incubated overnight at 37°C. A sterile 96-well Singer pin was used to pick up cells from each colony and inoculate 200 μL of LB in a 96-well plate. The parent-GFP strain was inoculated in some of the empty wells on the edge of each plate as controls. The plate was covered with an AeraSeal breathable film (Sigma-Aldrich) and grown on a plate shaker at 37°C for 4–5 hr until all wells were cloudy (OD$_{600}$ ~1.0).

One hundred microliters of each culture were pipetted into a separate 96-well plate with 100 μL of a parent-RFP culture in each well. This plate was used as the inoculum for the competitive fitness screen. The remainder of the cultures in the original 96-well plate were used as the inoculum for the monoculture screen. To quantify competitive fitness, a titration row of parent-RFP and parent-GFP mixtures in 10% increments (100% parent-RFP+0% parent-GFP; 90% parent-RFP+10% parent-GFP, etc.) was added to each plate (*Figure 1D*). Since the oxygen limitation that results in stationary cultures causes cell death in *B. subtilis* (*Arjes et al., 2020*), we ensured that the library was aliquoted and spotted within 1 hr. For most assays, a Singer ROTOR HDA pinning robot (Singer Instruments) was used to pin ~1 μL of liquid cultures onto LB or MSgg agar Singer PlusPlates (with 35 mL of medium poured on a level surface for co-cultures, or 50 mL of medium for monocultures), without and with xylose. We used the 'spot many' protocol of the Singer ROTOR HDA to mix the wells before spotting and transferred 12 times from the source liquid plate to the target agar plate. For some assays, a RAININ Benchsmart 96-well pipetting robot was used rather than the Singer ROTOR HDA to pipet 1 μL onto the agar plates. Agar plates were incubated at 30°C and placed in a box or were loosely covered in plastic to reduce drying.

When screen outliers (*Figures 2D* and *3C*) were replicated, strains were streaked for single colonies, which were inoculated into 200 μL of medium in the interior wells of a 96-well microtiter plate. The exterior wells were inoculated with the parent-GFP control strain, leaving the top for the parent-GFP+parent-RFP titration. To replicate findings regarding *alrA*, fresh colonies of the *alrA* knockdown strain and the parent-RFP strain were inoculated into 5 mL LB in test tubes and cultured on a roller drum at 37°C to an OD$_{600}$ ~1.0. Equal-volume mixtures of the cultures were spotted along with a parent-GFP+parent-RFP titration in 12- or 6-well plates.

## Imaging and image analysis of monoculture colonies in the library
Monoculture colonies were imaged using a Canon EOS Rebel T5i EF-S with a Canon Ef-S 60 mm f/2.8 Macro USM fixed lens. The DSLR camera was set up at a fixed height in a light box with diffuse lighting from three sides. The lighting and camera settings were maintained for the duration of the experiment using the 'manual' mode on the camera. The EOS Utility software was used to run the camera. Plates were imaged colony side up to avoid imaging through the agar. Images were analyzed using FIJI and scored as 'grew outside original spot,' 'did not grow outside original spot,' or 'died and/or threw off suppressors.' The ones classified as 'died and/or threw off suppressors' were assigned a colony size of 0. Suppressors were identified based on off-center colonies, often in flower petal-like arrangements in which one or a few cells within the original spot eventually grew but the majority of the cells did not. Colony size was measured manually in FIJI by drawing a diagonal line across the diameter of the colony.

## Imaging and image analysis of biofilm colony co-cultures
A Typhoon FLA 9500 scanner was used to image colonies using the multi-plate drawer. We used ~35 mL of medium with agar on a Singer rectangular plate to be near the plane of focus when imaging through agar. GFP (473 nm laser, Long Pass Blue filter) and RFP (532 nm laser, Long Pass Green filter) signals were acquired.

For image analysis, Typhoon RFP and GFP images were cropped to contain only one plate and the image was rotated so that A1 was in the top left corner. Custom Matlab code was used to read in the image of each plate, divide it into a grid in which each grid cell contained one colony, and extract the fluorescence level across the grid. The ratio of the extracted GFP and RFP values was computed for every colony, and the ratio values for the titration row against the fraction GFP were fitted using the function $I=\alpha G/(1–\beta G)$, where $I$ is the GFP/RFP ratio, $G$ is the fraction GFP, and $\alpha$ and $\beta$ are fit parameters (*Figure 1D*). The fit parameters from the titration row were used to map the library data and assign GFP fractions; the data were normalized so that the average of the internal

parent-RFP and parent-GFP co-culture controls was 1. The image analysis and GFP/RFP scripts are available on Data Dryad: doi:10.5061/dryad.79cnp5htm (*Arjes, 2021*).

## Harvesting cells to measure colony-forming units

Parent-RFP and parent-GFP were cultured and mixed in ratios from 0% to 100% in 20% increments as described above. One microliter of these mixtures was spotted onto LB or MSgg agar. Plates were incubated at 30°C in an unsealed ziplock bag with a wet paper towel. After 24 hr, colonies were imaged to quantify RFP and GFP fluorescence as described above. Individual colonies were scraped into 1 mL of 1× phosphate buffered saline using a sterile stick. Colonies were then sonicated using a Vibra-Cell sonicator (Sonics and Materials, Inc, Newtown, CT, USA) using cycles consisting of 12 1-s pulses with 20% amplitude. LB colonies were sonicated with 1–2 cycles, and MSgg colonies were sonicated with 5–7 cycles to disrupt the matrix and liberate individual cells. Cultures were visualized under a microscope to ensure that all cell clumps had been disrupted. Cell suspensions were serially diluted 10-fold and plated on LB. After overnight incubation at 37°C, plates were imaged to identify RFP- and GFP-expressing colonies and colony-forming units per mL were calculated.

## CRISPRi *rfp* knockdown

Wild-type 3610, parent-RFP, and CRISPRi-RFP strains were cultured in 5 mL test tubes at 37°C to an $OD_{600}$ ~1 in liquid LB. The parent-RFP strain was spotted onto LB and MSgg agar plates without xylose, while the CRISPRi-RFP strain was spotted onto LB and MSgg agar in 12-well plates containing 0.0005–1% xylose. RFP fluorescence of the colonies was imaged as described above, and FIJI was used to quantify the fluorescence intensity of each colony, using wild-type 3610 as a blank.

## Wild-type 3610, parent-GFP, and parent-RFP biofilm and non-biofilm colony growth

Wild-type 3610, parent-GFP, and parent-RFP strains were cultured in liquid LB to an $OD_{600}$ ~1 at 37°C. One microliter of each culture was spotted onto LB or MSgg agar in a six-well plate. Colonies were cultured for 48 hr at 30°C in an unsealed ziplock plastic bag with a wet paper towel to increase humidity. Colonies were imaged using the DSLR setup described above.

## Liquid culturing for growth rate analysis

A single colony was used to inoculate 200 µL LB in a 96-well microtiter plate. $OD_{600}$ was measured every 7.5 min using a Biotek Epoch plate reader at 37°C. $OD_{600}$ curves were blanked and smoothed. The maximum growth rate of each culture was defined as the maximum derivative of $\ln(OD_{600})$. The script used for analyzing the growth curves and calculating growth rates is available on Data Dryad: doi:10.5061/dryad.79cnp5htm (*Arjes, 2021*).

## Model of nutrient-dependent colony growth

To determine how competitive fitness in a co-culture colony is affected by differences in growth rate, we simulated a reaction-diffusion model in which two cell types with densities ($C_i$, $i = 1, 2$) are inoculated in a circular spot from which they spread randomly in two dimensions to compete for fresh nutrients ($n$) and grow with distinct maximal growth rates ($M_i$), according to the following equations:

$$\frac{\partial C_i}{\partial t} = M_i \frac{n}{n+K} C_i + D_C \nabla^2 C_i$$

$$\frac{dn}{dt} = -b \sum_i M_i \frac{n}{n+K} C_i + D_n \nabla^2 n,$$

where $D_C$ is the cell diffusivity, $D_n$ is the diffusivity of nutrients, $b$ is a conversion factor dictating how nutrients lead to cell growth, and cell growth is limited by nutrients when $n \sim K$ or less. Initially $n = n_0$ everywhere and, to represent the initial pinning of cells to the agar surface, $C_i = C_0$ within a disc of radius $r_0$ and outside of this disc $C_i = 0$. The transformations $\bar{n} = \frac{n}{n_0}$, $\bar{C}_i = \frac{C_i}{n_0/b}$ express the nutrient level normalized by its initial value (so $\bar{n}_0 = 1$) and $C_i$ normalized by its value if cells of type $i$

consume all initially available nutrients, respectively, and eliminate the parameter $b$ from the equations:

$$\frac{\partial \bar{C}_1}{\partial t} = M_1 \frac{\bar{n}}{\bar{n} + K/n_0} \bar{C}_1 + D_c \nabla^2 \bar{C}_1, \frac{\partial \bar{C}_2}{\partial t} = M_2 \frac{\bar{n}}{\bar{n} + K/n_0} \bar{C}_2 + D_c \nabla^2 \bar{C}_2$$

$$\frac{d\bar{n}}{dt} = -M_1 \frac{\bar{n}}{\bar{n} + K/n_0} \bar{C}_1 - M_2 \frac{\bar{n}}{\bar{n} + K/n_0} \bar{C}_2 + D_n \nabla^2 \bar{n}.$$

Four of the seven parameters were approximated from data. The maximal growth rate $M_1 = 0.0175$ min$^{-1}$ set the timescale and corresponds to a 40-min doubling time, similar to *B. subtilis* 3610 at 30°C. The radius of the initial spot ($r_0 = 1$ mm) set the spatial scale. The ratio of maximal growth rates $\frac{M_2}{M_1} = 0.8$ was set to match the ratio in LB for ribosomal knockdowns compared with wild type (*Figure 2C*). We estimated the initial areal density of cells compared with their saturation density to be $\frac{C_0}{n_0/b} \approx 10^{-3}$ within the initial spot. To obtain this estimate, we assumed 1 μL of stationary phase culture spotted $10^6$ cells over $\pi$ mm$^2$, and that the spotted cells saturate at a density of $10^9$ cells/mm$^2$ (assuming 1 mL of stationary phase culture contains $10^9$ cells [*Moran et al., 2010*] and concentrates into 1 mm$^3$ when pelleted, the latter indicating a maximal density of $10^9$ cells/mm$^3$ within a colony with height 1 mm).

The remaining three parameters ($D_C$, $\frac{K}{n_0}$, $\frac{D_n}{D_C}$) were fitted to data. The model approximately recapitulates the competitive fitness data (defined in the model as the ratio of integrated cell densities $\int C_2 dA / \int C_1 dA$) and colony radius at 16 hr when $D_C \approx 0.003$ mm$^2$ hr$^{-1}$ (*Figure 2E*, inset), while results were fairly robust over 100-fold ranges to variations in $\frac{K}{n_0} \approx 0.05$ and $\frac{D_n}{D_C} \approx 100$ (*Figure 2—figure supplement 1C*).

## DAVID functional enrichment

We used the DAVID functional annotation tool (https://david.ncifcrf.gov) to determine whether particular gene classes were enriched for each phenotype. The BSU identification numbers of the strains identified by our analysis and of the entire CRISPRi library were used as the 'list' and the 'background,' respectively, using the 'locus tag' option on the website.

## D-amino acid rescue experiments

The unlabeled *alrA* knockdown (HA420) and wild-type 3610 were grown to an OD$_{600}$ ~1 in liquid LB. One microliter of each culture was spotted onto MSgg or MSgg-xylose agar plates with 0.04 mg/mL of one of the D-amino acids. Cultures were incubated for 48 hr, and imaged at 24 and 48 hr using the DSLR setup described above.

## Liquid growth of *alrA* monocultures and co-cultures for plating efficiency

Cultures of the *alrA* knockdown strain (HA420) and the parent strain (HA2) were separately cultured from a colony in liquid LB at 37°C until both strains reached OD$_{600}$ ~1.0. HA420 and HA2 cultures were mixed 1:1, and the mixture along with the HA420 monoculture was back-diluted 1:200 into 3 mL MSgg medium with 1% xylose and incubated at 30°C, shaking at an angle. At 0, 1, 2, 3, 4, and 5 hr, cultures were sampled, 10-fold serially diluted, and spotted onto MLS or chloramphenicol selection plates to determine CFU/mL of each strain (HA2 and HA420 are MLS$^R$, HA420 is cm$^R$, HA2 is cm$^S$). We incubated dilutions overnight and counted colonies to calculate CFU/mL.

## Liquid growth of wild-type 3610 and *alrA* knockdowns for microscopy

The 3610 wild-type strain and an unlabeled *alrA* knockdown strain (HA420) were grown to an OD$_{600}$ ~1 in LB. Each strain was diluted 1:200 into 3 mL MSgg+1% xylose to fully knock down *alrA* transcription during incubation at 30°C, shaking at an angle. At 0 and 6 hr, 1 μL of each culture were spotted onto 1× PBS pads made with 1.5% agar. When dry, we added a coverslip and imaged the cells in phase contrast on a Nikon Ti-E inverted microscope using a ×100 objective (NA: 1.4).

## Two-dimensional culturing

Strains were grown in LB to $OD_{600}\sim1$. While strains grew, we prepared a large agar pad at least 1 hr before imaging using the bottom of a rectangular Singer PlusPlate culture plate and 30 mL of MSgg +1% xylose. After the agar solidified, we added a second Singer PlusPlate on top to prevent contamination and drying. We mixed the strains 1:1 volumetrically and spotted 0.5 µL of this mixture onto the agar pad. After the spot dried, we added a large 113 × 77 mm custom no. 1.5 glass coverslip (Nexterion, Jena, Germany). Pads were incubated in a plastic bag with a wet paper towel to maintain humidity at 30°C for 24 hr. The entire spot was captured in a grid of images using a Nikon Ti-E inverted microscope with a 40 air objective (NA: 0.95) integrated with µManager (*Edelstein et al., 2014*). Images were stitched together using custom Matlab (MathWorks, Santa Clara, CA, USA) code available on Data Dryad: doi:10.5061/dryad.79cnp5htm (*Arjes, 2021*). GFP and RFP stitched images were merged using Adobe Photoshop, adjusting each channel identically across all images.

## Mutualism screen

The mutualism screen (*Figure 5D*) was performed as described above, except two screens were performed side by side: one in which each strain in the *sacA::gfp* library was co-cultured with the parent-RFP strain (HA12), and one in which each strain was co-cultured with the *ΔepsH ΔtasA* parent-RFP strain (HA825). These screens were carried out on MSgg+1% xylose plates with a titration row of HA773 (parent-GFP) in combination with either HA12 or HA825, as described above.

## Qualitative fitness determination via dilution streaking

Strains were inoculated from a fresh colony into 5 mL LB and incubated at 37°C for ~5 hr on a roller drum. Cultures were streaked onto agar plates using sterile wooden sticks. A new sterile stick was used for each streak. Plates were incubated overnight (~18 hr) at 37°C and imaged using the DSLR camera setup described above.

## Statistical methods

All statistical tests are stated in the figure legends. To determine whether data were significantly different, Student's unpaired *t*-tests were applied. The Benjamini–Hochberg multiple-hypothesis correction was applied to data in *Figures 2C*, *3C,* and *5B*.

# Acknowledgements

The authors thank the Huang lab and Petra Levin for helpful discussions, Nicola Stanley-Wall for providing the *sacA::gfp* construct, and Dan Kearns for providing matrix mutant constructs. The authors also thank Ákos Kovács and Anna Dragos for experimental advice and Andrés Aranda-Díaz for providing help with coding and statistical analyses. The authors acknowledge support from the Allen Discovery Center at Stanford on Systems Modeling of Infection (to HAA and KCH), the Stanford Bioengineering Summer Research Experience for Undergraduates Program (to HG), and NIH K22 Award AI137122 (to JP). KCH is a Chan Zuckerberg Biohub Investigator.

# Additional information

### Funding

| Funder | Grant reference number | Author |
| --- | --- | --- |
| Paul G. Allen Family Foundation | Discovery Center at Stanford on Systems Modeling of Infection | Heidi A Arjes<br>Kerwyn Casey Huang |
| National Institutes of Health | K22 Award AI137122 | Jason Peters |
| Stanford Engineering | Research experience for undergraduates | Haiwen Gui |
| Chan Zuckerberg Biohub Investigator | | Kerwyn Casey Huang |

The funders had no role in study design, data collection and interpretation, or the decision to submit the work for publication.

### Author contributions
Heidi A Arjes, Conceptualization, Resources, Data curation, Software, Formal analysis, Supervision, Validation, Investigation, Visualization, Methodology, Writing - original draft, Writing - review and editing; Lisa Willis, Software, Validation, Investigation, Visualization, Methodology, Writing - original draft, Writing - review and editing; Haiwen Gui, Yangbo Xiao, Data curation, Investigation; Jason Peters, Conceptualization, Resources, Methodology, Writing - review and editing; Carol Gross, Conceptualization, Resources, Supervision, Methodology, Writing - review and editing; Kerwyn Casey Huang, Conceptualization, Resources, Software, Supervision, Funding acquisition, Methodology, Writing - original draft, Project administration, Writing - review and editing

### Author ORCIDs
Heidi A Arjes (iD) https://orcid.org/0000-0002-4708-4286
Haiwen Gui (iD) http://orcid.org/0000-0003-0564-940X
Carol Gross (iD) http://orcid.org/0000-0002-5595-9732
Kerwyn Casey Huang (iD) https://orcid.org/0000-0002-8043-8138

### Decision letter and Author response
Decision letter https://doi.org/10.7554/eLife.64145.sa1
Author response https://doi.org/10.7554/eLife.64145.sa2

## Additional files

### Supplementary files
- Supplementary file 1. Strains used in this study.
- Supplementary file 2. All data from systems-level competitive fitness screens.
- Supplementary file 3. Gene knockdowns that have different competitive fitness in LB versus MSgg.
- Supplementary file 4. Competitive fitness data of select gene knockdowns with media additives.
- Supplementary file 5. A list of gene knockdowns that display mutualism.
- Transparent reporting form

### Data availability
Related scripts and data deposited in Dryad Digital Repository (doi:10.5061/dryad.79cnp5htm). Remaining data generated or analysed during this study are included in the manuscript and supporting files.

The following dataset was generated:

| Author(s) | Year | Dataset title | Dataset URL | Database and Identifier |
|---|---|---|---|---|
| Arjes H | 2021 | Three-dimensional biofilm growth supports a mutualism involving matrix and nutrient sharing - related scripts and data | https://doi.org/10.5061/dryad.79cnp5htm | Dryad Digital Repository, 10.5061/dryad.79cnp5htm |

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
