## [Decision Letter]

**Acceptance summary:**

The authors have used tunable CRISPRi variants of *B. subtilis* to examine how different genes contribute to the interactions of cells within a biofilm. They observed that several genes have different fitness effects in biofilm colonies compared to no-biofilm colonies, as well as in monoculture versus co-culture. Interestingly they identified knockdowns with impaired growth in liquid cultures that were rescued in the context of an ancestral biofilm. Their findings add to our understanding of the presence and consequences of intercellular interactions within biofilm colonies.

**Decision letter after peer review:**

Thank you for submitting your article "Three-dimensional biofilm growth supports a mutualism involving matrix and nutrient sharing" for consideration by *eLife*. Your article has been reviewed by three peer reviewers, including Babak Momeni as the Reviewing Editor and Reviewer #1, and the evaluation has been overseen by Aleksandra Walczak as the Senior Editor. The following individual involved in review of your submission has agreed to reveal their identity: Mariana Gomez Schiavon (Reviewer #3).

The reviewers have discussed the reviews with one another and the Reviewing Editor has drafted this decision to help you prepare a revised submission.

Overall, the reviewers all agreed that the submitted manuscript is substantial, well-explained, and a nice addition to the field. They agreed that in principle this work is suitable for publication in *eLife*.

Revisions:

The main concern raised by the reviewers is the lack of mechanistic insight, when the "matrix-dependent mutualism" is discussed in the Results section. Is the complementation and synergism really dependent on growth in biofilms, or are these caused due to the cultivating conditions (such colonies have extremely reduced oxygen levels inside, for instance)? What is missing in this current manuscript is nicely summarized by the authors: "some aspect of three-dimensional growth beyond matrix production is fundamental to the rescue." This aspect could have been explored more by using microscopy to show the distribution of different genotypes. The reviewers identified this as a weakness. At the same time, the reviewers are cognizant of the substantial contribution of this manuscript and current COVID limitations in running experiments. One solution is to: (A) perform the synergism assays in air-liquid biofilms. If these shows that similar synergism exists, the authors can claim that these observations are due to the 3D environment of the biofilms. (B) if the experiments with air-liquid biofilm do not support their hypothesis, or the authors decides not to perform these experiments, in this case, their claim connected to biofilm and 3D context should be removed and claim only "growth in a colony on an agar medium". In the latter case, the reviewer recommend revising the phrasing in the Results section to leave room for alternative explanations and expand the issue in the Discussion section to include hypotheses about what the mechanistic reason might be.

Another issue raised by one of the reviewers is a concern about the experimental protocol: none of the fluorescence-based fitness measurements are validated with CFU assays. Importantly, all colony measurement has been performed based on fluorescence, while liquid cultures were assayed using survival (CFU). Because the spatial organization is not examined in these colonies, it is not possible to confirm the current interpretation of the data. In the current form, one cannot distinguish whether fitness differences exist or only positioning in the colonies are altered by metabolic differences or lack vs presence of the matrix (e.g., see https://doi.org/10.1371/journal.pcbi.1000716 and https://doi.org/10.1073/pnas.1323632111). The colonies should be also assayed as described in L330 to 336 of planktonic cultures. The reviewers recommend that the authors include additional evidence that would justify their approach.

[Editors' note: further revisions were suggested prior to acceptance, as described below.]

Thank you for resubmitting your work entitled "Three-dimensional biofilm growth supports a mutualism involving matrix and nutrient sharing" for further consideration by *eLife*. Your revised article has been evaluated by Aleksandra Walczak (Senior Editor) and a Reviewing Editor.

Summary:

The manuscript has been improved and the majority of reviewers comments have been addressed. However, there are some remaining issues that need to be addressed before acceptance, as outlined below:

After consulting with the reviewers, the only remaining issue is the validation of the results between liquid and spatial contexts. One of the reviewers had requested a validation of colony CFUs and pellicle biofilm results in the previous round and they did not find the changes satisfactory. All reviewers agreed that if the authors decide not to validate the observation from colonies on other biofilm systems, the issue can be addressed by changing the language from "biofilm" to "colony" or "community" throughout the paper (including the title). The authors have already done this in a limited scale in their last revision, but the reviewer expects the change throughout the manuscript. We also suggest a minor revision in the “Media” subsection: please correct to "note that original description of the MSgg medium includes KOH for adjusting the pH, thus it does not contain Na+".

---

## [Author Response]

Revisions:The main concern raised by the reviewers is the lack of mechanistic insight, when the "matrix-dependent mutualism" is discussed in the Results section. Is the complementation and synergism really dependent on growth in biofilms, or are these caused due to the cultivating conditions (such colonies have extremely reduced oxygen levels inside, for instance)? What is missing in this current manuscript is nicely summarized by the authors: "some aspect of three-dimensional growth beyond matrix production is fundamental to the rescue." This aspect could have been explored more by using microscopy to show the distribution of different genotypes. The reviewers identified this as a weakness. At the same time, the reviewers are cognizant of the substantial contribution of this manuscript and current COVID limitations in running experiments. One solution is to: (A) perform the synergism assays in air-liquid biofilms. If these shows that similar synergism exists, the authors can claim that these observations are due to the 3D environment of the biofilms. (B) if the experiments with air-liquid biofilm do not support their hypothesis, or the authors decides not to perform these experiments, in this case, their claim connected to biofilm and 3D context should be removed and claim only "growth in a colony on an agar medium". In the latter case, the reviewer recommend revising the phrasing in the Results section to leave room for alternative explanations and expand the issue in the Discussion section to include hypotheses about what the mechanistic reason might be.

Thank you for these suggestions. First, the reviewers suggest that “This aspect could have been explored more by using microscopy to show the distribution of different genotypes.” In the text, we have now expanded on our data regarding the spatial distributions of phenotypes within colonies, which suggest that matrix-capable mutants support growth of the matrix-deficient parent strain (see Figure 5A).

Second, air-liquid biofilms are tricky to achieve with our mutants, as we cannot ensure that the biofilm forms and is stable enough before essential-gene depletions die. This issue is especially the case for *alrA* depletion, as *alrA* knockdown results in cell death within hours in liquid cultures (Figure 4E). Moreover, any suppressors present at the onset of the experiment would grow and spread, making it impossible to tell whether GFP signal is from suppressors or non-suppressor mutants; suppressors are obvious on a surface due to petal-like colonies arising from single mutants in the initial spot. As the reviewers suggested, we have modified the claim regarding biofilms and three-dimensional context to state “three-dimensional colony”.

Lastly, the reference to “mechanistic insights” is not totally clear to us; we assume it relates to the precise nature of how spatial structure dictates survival. While there is certainly more to be discovered in this regard, we would like to point out that we show that the mechanism of mutualism between matrix-proficient *alrA* cells and matrix-deficient parent cells is due to D-alanine sharing (from the parent to *alrA*) and matrix sharing (from the *alrA* depletion to the parent). The spatial distribution of both mutants throughout the entire colony suggests that matrix is shared between the strains (since the strains are not spatially segregated, Figure 5A). We suspect that some other mutualism candidates in Figure 5E also involve nutrient/matrix sharing mechanisms, especially the menaquinone synthesis and glutamate racemase genes. It remains unclear whether all the other genes identified in the screen (*scpA*, *hbs*, *rnz*, *frr*) involve nutrient sharing or another as of yet unidentified mechanism that is beyond the scope of the current manuscript. At the reviewer’s suggestions, we have clarified and expanded the appropriate Results and Discussion sections.

Another issue raised by one of the reviewers is a concern about the experimental protocol: none of the fluorescence-based fitness measurements are validated with CFU assays. Importantly, all colony measurement has been performed based on fluorescence, while liquid cultures were assayed using survival (CFU). Because the spatial organization is not examined in these colonies, it is not possible to confirm the current interpretation of the data. In the current form, one cannot distinguish whether fitness differences exist or only positioning in the colonies are altered by metabolic differences or lack vs presence of the matrix (e.g., see https://doi.org/10.1371/journal.pcbi.1000716 and https://doi.org/10.1073/pnas.1323632111). The colonies should be also assayed as described in L330 to 336 of planktonic cultures. The reviewers recommend that the authors include additional evidence that would justify their approach.

We apologize for the confusion, we have investigated the spatial organization for the *alrA* mutants as well as the other mutualism candidates (see Figure 4A, 5A, 5E). When matrix production was normal, *alrA* mutants grew to fill the initial inoculum and were able to grow radially outward from the initial spot but were eventually outcompeted by parent-RFP (Figure 5A). By contrast, when the *alrA* mutant was the sole provider of matrix, the spatial organization was highly different, with both *alrA* cells and the parent occupying the entire colony, suggesting that extracellular matrix was shared between these strains. We have clarified these points in the Results section.

We appreciate the reviewer’s suggestion to validate our fluorescence-based data with CFU counts. To obtain accurate CFU counts from a colony, it is necessary to resuspend the colony and sonicate to disrupt the biofilm matrix and liberate individual cells. We were concerned that *alrA*-depletion strains would be more sensitive to sonication as we demonstrated that the mutation disrupts the cell wall, so we instead validated our fluorescence assay with CFU counting of different mixtures of the parent-GFP and parent-RFP strains. Both CFU counting and colony fluorescence methods mapped well to our predicted ratio function fit line, and thus colony fluorescence is validated as a useful and more efficient tool to compare competition in colonies in high throughput (new Figure 1—figure supplement 1B). CFU counting is the current standard and widely used assay for measuring competition in liquid cultures, and that is why we used it for our liquid co-culture measurements. We did not verify our liquid experiments with fluorescence because the cultures were highly dilute in the initial few hours when our CFU data indicated that cells die, and hence the fluorescence signals would be too weak to compare to our CFU data.

[Editors' note: further revisions were suggested prior to acceptance, as described below.]

The manuscript has been improved and the majority of reviewers comments have been addressed. However, there are some remaining issues that need to be addressed before acceptance, as outlined below:After consulting with the reviewers, the only remaining issue is the validation of the results between liquid and spatial contexts. One of the reviewers had requested a validation of colony CFUs and pellicle biofilm results in the previous round and they did not find the changes satisfactory. All reviewers agreed that if the authors decide not to validate the observation from colonies on other biofilm systems, the issue can be addressed by changing the language from "biofilm" to "colony" or "community" throughout the paper (including the title). The authors have already done this in a limited scale in their last revision, but the reviewer expects the change throughout the manuscript.

We appreciate the reviewer’s desire for specificity, and recognize that some of the biofilm field view pellicles and biofilm colonies to be substantively different in biofilm context. We have endeavored to change the text in a way that will satisfy the reviewer, but still maintain clarity about our results, as we describe below – we are not tied to any of the terminology, and if there are other options that would be acceptable to the reviewer, we would be happy to further modify the text.

In terms of clarity, we feel that to change all instances of biofilm simply to “colony” would create confusion, as many of our results are about the distinction between MSgg and LB colonies, and we chose the media specifically because of their promotion of what are generally thought of as biofilms, or lack thereof. Indeed, colonies on MSgg have long been referred to as biofilms or biofilm colonies; in the book Biofilm Development with an Emphasis on *Bacillus subtilis*, Roberto Kolter and colleagues write on page 3 of Chapter 1 that “the colonies that grow on the surface of agar dishes and demonstrate macroscopically complex architectures are now widely recognized as a form of biofilm (reviewed in Branda et al., 2005).” We considered changing the terminology to “MSgg colonies” and “LB colonies,” but such terminology would be inscrutable in the title and abstract and also feel that this would make the manuscript difficult to read for anyone not an expert in the *B. subtilis* biofilm field.

Thus, we have ensured that throughout our manuscript, all mentions of biofilms have been changed to “biofilm colony” rather than “biofilms,” which we hope the reviewer will feel is clearly distinguished between the “non-biofilm colonies” grown on LB and “biofilm colonies” grown on MSgg. We have made sure that this point is emphasized in the manuscript, as we write:

“However, it is important to note that the biofilm definition is nuanced; colonies on LB may have some biofilm characteristics [10], and biofilm pellicles formed at air–liquid interfaces may have distinct properties from MSgg colonies.”

(It appears to be standard for the field to reserve the use of “community” for when there is a stable culture of multiple species, so we avoided this term.)

We also suggest a minor revision in the “Media” subsection: please correct to "note that original description of the MSgg medium includes KOH for adjusting the pH, thus it does not contain Na+".

We would like to satisfy the reviewer (who noted that these details have no impact on the results in our study), but we were unable to confirm the validity of their statement about the original description including KOH for adjusting the pH. When searching for the original description for this, we found this recipe for MSgg in Branda et al., 2001 “Fruiting body formation by *Bacillus subtilis*”:

“Minimal medium (MSgg): 5 mM potassium phosphate (pH 7)/100 mM Mops (pH 7)/2 mM MgCl_2_/700 μM CaCl_2_/50 μM MnCl_2_/50 μM FeCl_3_/1 μM ZnCl_2_/2 μM thiamine/0.5% glycerol/0.5% glutamate/50 μg/ml tryptophan/50 μg/ml phenylalanine (adapted from ref. 10)”

Reference 10 is: “Partial Purine Deprivation Causes Sporulation of *Bacillus subtilis* in the Presence of Excess Ammonia, Glucose, and Phosphate” by E. Freese, J.E. Heinze, and E.M. Galliers (1979).

They state:

“Media. A synthetic medium was used which contained 10 mM (NH_4_)_2_SO_4_, 5 mM potassium phosphate (pH 7.0), 100 mM morpholinopropanesulphonate (adjusted to pH 7 with NaOH), 2 mM MgCl_2_, 0.7 mM CaCl_2_, 50 μM MnCl_2_, 5 μM FeCl_3_, 1 μM ZnCl_2_, 2 μM thiamin, and the stated amounts of D-glucose and L-glutamate (adjusted to pH 7.0 with KOH) as carbon sources. (*Bacillus subtilis* metabolizes glucose rapidly and glutamate very slowly.) Added to this medium were the compounds required by auxotrophs (each at 50 μg ml^-1^) except where otherwise stated. Adenine, AICA, guanine, guanosine, hypoxanthine and xanthine stock solutions were prepared in dilute KOH.”

Therefore, it appears the original media used in Branda et al., 2001 contains Na^+^, thus it would be improper to state otherwise in our manuscript.

We additionally checked a handful of papers from the Losick, Kolter, and Shank laboratories and there is no mention of what they used to adjust the pH of their MOPS buffer, so it is unclear if they use KOH or NaOH, and thus the statement stands as written.

Therefore, we think that the text in our revised manuscript (“note that some laboratories adjust the pH of MSgg with KOH”) is sufficient. If the reviewer can provide a reference to the recipe of MSgg that states the authors use KOH, we would be happy to include it as a reference. (To be entirely clear, we replaced “MSgg” with “MOPS” buffer in this line in our new revision, as the MOPS buffer is what is adjusted for pH.)